

# Growth regulators promote soybean productivity: a review

Hanna Amoanimaa-Dede[1], Chuntao Su[1], Akwasi Yeboah[1], Hang Zhou[1], Dianfeng Zheng[1,2] and Hongbo Zhu[1]

[1] College of Coastal Agricultural Sciences, Guangdong Ocean University, Zhanjiang, Guangdong Province, China
[2] Shenzhen Institute of Guangdong Ocean University, Shenzhen, Guangdong Province, China

## ABSTRACT

Soybean [*Glycine max* (L.) Merrill] is a predominant edible plant and a major supply of plant protein worldwide. Global demand for soybean keeps increasing as its seeds provide essential proteins, oil, and nutraceuticals. In a quest to meet heightened demands for soybean, it has become essential to introduce agro-technical methods that promote adaptability to complex environments, improve soybean resistance to abiotic stress , and increase productivity. Plant growth regulators are mainly exploited to achieve this due to their crucial roles in plant growth and development. Increasing research suggests the influence of plant growth regulators on soybean growth and development, yield, quality, and abiotic stress responses. In an attempt to expatiate on the topic, current knowledge, and possible applications of plant growth regulators that improve growth and yield have been reviewed and discussed. Notably, the application of plant growth regulators in their appropriate concentrations at suitable growth periods relieves abiotic stress thereby increasing the yield and yield components of soybean. Moreover, the regulation effects of different growth regulators on the morphology, physiology, and yield quality of soybean are discoursed in detail.

## INTRODUCTION

Grain legumes are vital and dominant group of crops after cereals such as wheat, rice, and maize which contribute immensely to human nutrition. Their nutritive value is appreciable especially to the developing world as these regions have inadequate supply of food derived from animal sources (*Shakya, Patel & Singh, 2016*). Among the legumes, soybean (*Glycine max* (L.) Merrill), is a unique and distinct crop belonging to the *Fabaceae* (*Leguminosae*) family, order *Fabales*, and sub-family *Papilionaceae* (*Pierozzi et al., 2008*). The high nutritional value of soybean makes it an indispensable crop, ranked 6th in terms of total crop yield and the most cultivated oilseed crop which adapts to varied climatic conditions (*Kim et al., 2015*). The major soybean producers in the world include USA, Brazil, Argentina, China, and India, contributing about 90% of global production (*Rizzo & Baroni, 2018*). Soybean possesses several vital nutrients such as carbohydrates, fats, proteins, vitamins, α-tocopherol, and minerals (Table 1). This food crop is widely consumed due to its excellent source of superior quality proteins. Lower levels of blood

Corresponding authors
Dianfeng Zheng,
zhengdf@gdou.edu.cn
Hongbo Zhu, tdzhu@gmail.com,
tdzhu@126.com

cholesterol, prevention of cardiovascular disease, breast cancer, osteoporosis in women, and relief of menopausal symptoms are some of the beneficial health effects derived from the consumption of soybean protein (*Gutiérrez et al., 2006*). Also, continuous consumption might efficiently control aging.

Generally, grain legumes are cultivated on marginal lands with less financial resources resulting in low yield. Genetic improvement, fertilizer and pesticide application, coupled with better cultural practices have been employed in an attempt to increase the productivity of these crops (*Khan & Mazid, 2018*). However, particular considerations have not been directed towards the physiological processes which reduce crop productivity. Plant growth regulators (PGRs) could be effective in realizing the yield potentials of these crops due to their numerous effects on plant life including flowering, growth, ion transport, and fruiting (*Ali & Bano, 2008*). Again, PGRs regulate the expression of endogenous hormones, improve plant physiology metabolism, and increase crop yield (*Zeng et al., 2012*). In particular, PGRs can optimize the process of photosynthesis and play a substantial role in maximizing crop yields. PGRs enhance the transportation of photosynthates, augment the source–sink association and thus, improve productivity (*Khan & Mazid, 2018*). Although PGRs have pronounced potentials, their application and accumulation must be carefully designed with regards to the specific species, optimum concentration, climatic conditions, and the stage of growth. Effective application on every facet of plant growth could increase total productivity by 10–15 million tons annually (*Khan & Mazid, 2018*).

Plant growth regulators are known to increase yield and bring about the desired growth per unit land and time. PGRs are also essential for germination, flower and root development, seed maturation, storage, and other functional processes in plants (*Wu et al., 2017*; *Yi-Ping et al., 2015*). In soybean production, PGRs such as auxin, cytokinin, abscisic acid, ethylene, jasmonic acid, salicylic acid, gibberellins, and brassinosteroids among others have been reported to promote growth and productivity (*Basuchaudhuri, 2016*; *Dhakne et al., 2015*; *Giri et al., 2018*; *Kim et al., 2018*; *Mohamed & Latif, 2017*; *Roy Choudhury, Johns & Pandey, 2019*; *Sudadi & Suryono, 2015*). PGRs sustainably promote root and shoot growth (*Kim et al., 2018*; *Qi et al., 2013*; *Steffens, Wang & Sauter, 2006*), enhance water use efficiency (*Giri et al., 2018*), promote flowering and pod setting (*Nagel et al., 2001*), increase chlorophyll content (*Sun et al., 2016*), improve photosynthetic rate (*Qi et al., 2013*; *Travaglia, Reinoso & Bottini, 2009*), enhance the translocation of photoassimilates (*Liu et al., 2019*; *Sun et al., 2016*), increase biomass accumulation (*Liu et al., 2019*; *Mohamed & Latif, 2017*), and induce tolerance to several abiotic stresses (*George, Hall & De Klerk, 2008*; *Hamayun et al., 2010*), resulting in enhanced growth and yield.

No comprehensive review is available on the effects of growth regulators on the growth and development of the soybean plant. Hence, we decipher current knowledge and possible applications of plant growth regulators in improving growth and yield. Notably, the regulation effects of various PGRs on the physiological, morphological, and biochemical properties as well as yield and quality traits of soybean are discussed in detail. Also, the mitigation effect of PGRs on various abiotic stress conditions to promote the growth and productivity of soybean are conferred. Based on the reviewed studies, we provide possible insights into the current applications of different PGRs that may affect the morphology,

**Table 1 Nutritional value of soybean per 100 g of dry matter.** *Source* adapted from (*Lokuruka, 2010*).

| Nutrient | | Amount |
|---|---|---|
| **Energy** | | 1,866 kJ (446 kcal) |
| **Carbohydrates** | | 30.16 g |
| | Sugars | 7.33 g |
| | Dietary fiber | 9.3 g |
| **Fat** | | 19.94 g |
| | Saturated | 2.884 g |
| | Monounsaturated | 4.404 g |
| | Polyunsaturated | 11.255 g |
| | Omega 3 | 1.330 g |
| | Omega 6 | 9.925 g |
| **Protein** | | 36.49 h |
| | Tryptophan | 0.591 g |
| | Threonine | 1.766 g |
| | Isoleucine | 1.971 g |
| | Leucine | 3.309 g |
| | Lysine | 2.706 g |
| | Methionine | 0.547 g |
| | Cystine | 0.655 g |
| | Phenylalanine | 2.122 g |
| | Tyrosine | 1.539 g |
| | Valine | 2.029 g |
| | Arginine | 3.153 g |
| | Histidine | 1.097 g |
| | Alanine | 1.915 g |
| | Aspartic acid | 5.112 g |
| | Glutamic acid | 7.874 g |
| | Glycine | 1.880 g |
| | Proline | 2.379 g |
| | Serine | 2.357 g |
| **Vitamins** | Vitamin A equivalent | 1 μg |
| | Thiamine (B1) | 0.874 mg |
| | Riboflavin (B2) | 0.87 mg |
| | Niacin (B3) | 1.623 mg |
| | Pantothenic acid (B5) | 0.793 mg |
| | Vitamin B6 | 0.377 mg |
| | Folate (B9) | 375 μg |
| | Choline | 115.9 mg |
| | Vitamin C | 6.0 mg |
| | Vitamin E | 0.85 mg |
| | Vitamin K | 47 μg |
| **Minerals** | Calcium | 277 mg |
| | Copper | 1.658 mg |

**Table 1** (*continued*)

| Nutrient | | Amount |
|---|---|---|
| | Iron | 15.7 mg |
| | Magnesium | 280 mg |
| | Manganese | 2.517 mg |
| | Phosphorus | 704 mg |
| | Potassium | 1797 mg |
| | Sodium | 2 mg |
| | Zinc | 4.89 mg |
| **Water** | | 8.45 g |

**Notes.**
KJ, Kilo-joules; kcal, kilocalorie; g, grams; mg, milligrams; $\mu$g, micrograms.

physiology, and yield quality of soybean. This article should be of particular interest to readers in the areas of plant physiology and breeding.

## SURVEY METHODOLOGY

An in-depth literature search of relevant academic articles in databases such as PubMed, Web of Science, Science Direct, and Google Scholar as well as the University's databases for books, journals, and reports was used for the compilation of this article. The search results were achieved through the use of phrases like ''growth regulators promote soybean growth'', ''PGRs affect soybean productivity'', PGRs influence soybean morphology'', ''PGRs mitigate abiotic stress in soybean'', etc. together with the use of ''NOT'', ''+'', ''AND'', ''vs'' for a particular search outcome. Generally, our search focused mainly on the effects of plant growth regulators in regulating the growth and development of the soybean plant and subsequently, increasing productivity. Since there has not been such a comprehensive compilation of studies on this topic, relevant related literature including those dating as far back as the late 1970s and 1980s were reviewed but we mainly focused on works from the past 10 years. Closed access articles and pieces of literature unavailable online that had relevant pieces of information were obtained through University's document delivery and inter-library loan services. Finally, the most relevant articles were used.

## MAJOR GROWTH REGULATORS

Plant growth regulators are carbon-based compounds either natural or artificial apart from nutrients, that perform fundamental roles in the life cycle of plants and alter or restore growth patterns of plants (*Davies, 2013*). Similar chemicals that affect plant growth are also produced by fungi and bacteria and can be used to control plant growth and development (*Khan & Mazid, 2018*). When these regulators are applied in lower quantities, they stimulate the natural systems that regulate growth to bring about rapid changes in the plant by either promoting or inhibiting plant growth from germination to senescence, thus, classified as growth promoters and growth retardants. They are commonly classified as phytohormones including; gibberellins, auxins, ethylene, cytokinins, abscisic acid, and synthetic substances that act like or against them. In recent times, other PGRs such as brassinosteroids, strigolactones, salicylic acid, polyamines, and jasmonic acid among others

have also been identified to induce physiological responses in plants (*Verma, Ravindran & Kumar, 2016*). In the majority of cases, PGRs affect the balance of plant hormones in treated plants. PGRs usually work together with other growth regulators and their effects overlap with each other. Table 2 summarizes the effects of different growth regulators on the growth and development of soybean as outlined by different authors.

## Auxins

Auxins are organic phytohormones with morphogen-like physiognomies implicated to coordinate several developmental processes in plants (*Strydhorst et al., 2017*). Synthetic auxins including 2, 4- dichloro-phenoxyaectic acid (2, 4-D), 2, 4, 5-trichloro-phenoxyacetic acid, Naphthalene acetic acid (NAA), 2-methyl-4-chloro-phenoxyaectic acid (MCPA), and Indole butyric acid (IBA) mimic the physiological responses of Indole acetic acid (IAA, the universal natural auxin) but are not as active as IAA (*Rademacher, 2015*). Auxins enhance physiological processes that directly control plant growth. For instance, low concentrations of auxins promote phototropism, lateral root initiation, gravitropism, vascular development, influence apical dominance (*Davies, 2010*; *Rademacher, 2015*), longitudinal shoot growth (*George, Hall & De Klerk, 2008*), root formation in cuttings (*Small & Degenhardt, 2018*), and regulate root growth by maintaining root stem cell niche (*Liu et al., 2017*). Exogenous IAA supplementation increased plant height, leaf number per plant, fruit size, and seed yield of groundnut, cotton, cowpea, and rice (*Kapgate et al., 1989*; *Kaur & Singh, 1987*; *Khalil & Mandurah, 1989*; *Lee, 1990*). However, high concentrations can negatively affect plants causing oxidative stress, cellular deaths (*Flasinski & Hac-Wydro, 2014*), and suppress axillary bud development (*Shimizu-Sato, Tanaka & Mori, 2009*). Thus, described as both stimulators and inhibitors of growth depending on the concentration (*Harms & Oplinger, 1988*).

## Gibberellins (GA)

Gibberellins are a major group of tetracyclic diterpenoid compounds with diverse properties known to influence various plant developmental processes. Gibberellic acid ($GA_3$) is the most produced and frequently used gibberellin associated with plant growth and development (*Gupta & Chakrabarty, 2013*). $GA_3$ promotes photosynthesis, seed germination, flowering, stem elongation, leaf growth, and cell division in plant shoot caused by the direct regulation of protein and RNA (ribonucleic acid) synthesis (*Hyun et al., 2016*). Also, gibberellins boost longitudinal growth caused by the development of meristematic tissues (*Rademacher, 2015*).

Exogenously applied $GA_3$ elevates the activities of some major enzymes such as ribulose-1, 5 bisphosphate carboxylase (Rubisco; *Yuan & Xu (2001)*, carbonic anhydrase, and nitrate reductase (*Aftab et al., 2010*). Recent reports revealed that GA controls some life processes in plants with regards to stress (*Hamayun et al., 2017*; *Wang et al., 2017*) and break dormancy in seeds (*Gupta & Chakrabarty, 2013*). Additionally, $GA_3$ can either reduce or inhibit the effects of water stress during germination and seedling emergence (*Kaur, Gupta & Kaur, 1998a*) and ameliorate the effects of salt stress, hence, maintaining standard growth and development (*Hamayun et al., 2010*).

**Table 2  Effects of different PGRs on soybean growth and development reported by different authors.**

| Name of PGR | Effect | Reference |
|---|---|---|
| Auxin | Improves nodulation | *Rademacher (2015), Roy Choudhury, Johns & Pandey (2019)* |
| | Development of shoot architecture | *Sarkar, Haque & Abdul Karim (2002)* |
| | Increases dry matter accumulation and seed yield | *Basuchaudhuri (2016), Dhakne et al. (2015)* |
| | Improves nitrate reductase activity | *Senthil, Pathmanaban & Srinivasan (2003)* |
| | Increases protein content of seeds | *Basuchaudhuri (2016)* |
| Gibberellins | Enhances water use efficiency (WUE) | *Giri et al. (2018)* |
| | Improves enzyme activity and photosynthesis | *Yuan & Xu (2001)* |
| | Induces nodulation | *Roy Choudhury, Johns & Pandey (2019)* |
| | Improves tolerance to abiotic stress | *Hamayun et al. (2017), Qin et al. (2011), Wang et al. (2017)* |
| | Promotes the development of adventitious roots | *Steffens, Wang & Sauter (2006)* |
| | Increases seed oil content | *Travaglia, Reinoso & Bottini (2009)* |
| | Increases yield and yield components | *Dhakne et al. (2015), Khatun et al. (2016), Upadhyay & Rajeev (2015)* |
| | Induces nodulation | *Roy Choudhury, Johns & Pandey (2019)* |
| | Increases flowering, pod setting and seed yield | *Nagel et al. (2001)* |
| Abscisic acid | Induces stress tolerance | *O'Brien & Benková (2013)* |
| | Enhances water use efficiency (WUE) | *Gao et al. (2016), Zhao et al. (2018)* |
| | Improves photosynthesis | *Qi et al. (2013)* |
| | Induces adventitious root formation | *Steffens, Wang & Sauter (2006)* |
| Ethylene | Promotes root and shoot growth, and increases root surface area | *Kim et al. (2018)* |
| | Increases tolerance to stress | *Hamayun et al. (2015)* |
| Salicylic acid | Increases chlorophyll content, photosynthesis, shoot biomass, and improves antioxidant enzyme activity | *Simaei et al. (2011)* |
| | Induces nodulation | *Roy Choudhury, Johns & Pandey (2019)* |
| | Improves vegetative growth and yield, increases protein and oil content | *Devi et al. (2011), Khatun et al. (2016)* |
| Jasmonic acid | Increases chlorophyll content | *Hassanein et al. (2009)* |
| | Increases biomass accumulation and grain yield, improves stress tolerance | *Mohamed & Latif (2017)* |
| | Increases nodulation | *Roy Choudhury, Johns & Pandey (2019)* |
| | Increases tolerance to stress, biomass accumulation, water use potential, translocation of photoassimilates, and productivity | *Zhang et al. (2004), Zhang et al. (2008)* |
| | Improves nodulation | *Sudadi (2012)* |
| Amine compounds | Enhances root elongation and increases endogenous hormone levels | *Qi et al. (2013)* |
| | Improves chlorophyll content, photosynthesis and $CO_2$ assimilation rate | *Qi et al. (2013), Sun et al. (2016), Liu et al. (2019)* |
| | Reduces abscission enzyme activity and increases antioxidant enzyme activity | *Sun et al. (2016)* |
| | Delays leaf senescence | *Zheng et al. (2008)* |
| | Increases dry matter accumulation and seed yield | *Qi et al. (2013), Liu et al. (2019)* |

## Cytokinins

Cytokinins are phytohormones synthesized in meristematic organs and tissues (*Rademacher, 2015*). Cytokinins regulate diverse physiological and biochemical processes in multiple plant organs, cell proliferation and differentiation, and plant response to stress conditions. The lack of cytokinin may halt the cell cycle and affect cellular activities (*George, Hall & De Klerk, 2008*) due to its cardinal role in plant cell division by directly regulating protein synthesis during mitosis. Cytokinins promote root growth, shoot development from internodes, chloroplast maturity, initiate callus formation (*Carrow & Duncan, 2011*; *George, Hall & De Klerk, 2008*) and stimulate plant responses to diverse biotic and abiotic stresses as well as nutrients by preventing cell degeneration, protein synthesis signaling, and augmenting protective enzymes (*Carrow & Duncan, 2011*). Exogenous cytokinin increased pod setting by inhibiting flower abortion in soybean and lupin and ultimately increased yield (*Najafian et al., 2009*). *Dietrich et al. (1995)* reported that the exogenous application of cytokinin to maize plants during pollination increased the number of kernels and the total kernel weight per ear by reducing apical kernel abortion. Benzyladenine (BA) application stimulated cell division and elongation, increased flower production, and decreased flower drop (*Krug et al., 2006*). Daily application of kinetin (Kn; for 5 d starting from 2 d post-anthesis) increased cell division and grain weight of rice (*Yang et al., 2003*).

## Abscisic acid (ABA)

Abscisic acid, a key regulator of abiotic stress resistance in plants (*Zhu et al., 2017*), also functions to coordinate several cardinal growth and developmental processes (*Wani & Kumar, 2015*). In plants, ABA is biosynthesized upon exposure to severe stress conditions to induce stress tolerance (*O'Brien & Benková, 2013*). Thus, referred to as the "stress hormone" (*Mehrotra et al., 2014*). ABA improves osmotic stress tolerance, protein and lipid synthesis, regulates protein synthesis genes, protects plants against pathogens, controls water and ion uptake, morphogenesis, and embryogenesis, breaks seed dormancy, and decreases leaf abscission (*Flasinski & Hac-Wydro, 2014*; *Strydhorst et al., 2017*). Exogenous ABA promotes primary root elongation and leaf senescence in plants. For instance, *Spollen et al. (2000)* reported that the application of ABA caused the primary root elongation of maize under low water potential. In *Arabidopsis*, ABA promoted leaf senescence and primary root elongation of seedlings (*Zhao et al., 2018*). Furthermore, exogenous ABA application in rice induced leaf yellowing, a major indicator of leaf senescence (*Fang et al., 2008*).

## Ethylene (ET)

Ethylene ($C_2H_4$) is a multifunctional hormone synthesized by practically all tissues connected with plant growth in the presence of oxygen (*Lin, Hsu & Wang, 2010*). Its effects on growth and development depend on the ABA, cytokinin, and auxin concentration, light, carbon dioxide, and the plant (*George, Hall & De Klerk, 2008*). Based on the plant species, concentration, and time of application, ET may promote, inhibit, or induce growth and senescence (*Pierik et al., 2006*). For example, low concentrations of ethephon (an ET releasing compound) increased the leaf area of mustard but no observed increase was

recorded at high concentrations (*Khan et al., 2008*). ET promotes the overall growth of plants by regulating secondary metabolites, cell division, cell size, fruit ripening, flowering, stolon formation, and root initiation (*George, Hall & De Klerk, 2008*; *Schaller, 2012*). ET in synergy with other growth regulators promotes plant growth and alleviates dormancy caused by environmental conditions (*Kepczynski & Van Staden, 2012*). For instance, high concentrations of cytokinin together with ET promoted root growth (*O'Brien & Benková, 2013*).

Research has established that ET promotes growth by increasing tolerance to diverse stress conditions. For instance, the application of ET improved tolerance under waterlogging conditions in rice by inducing the emergence of adventitious roots, aerenchyma formation in roots, and shoot elongation (*Ma, Chen & Zhang, 2010*). ET regulates abscisic acid and gibberellin homeostasis to stimulate growth (*Ahammed et al., 2020*) and also facilitates plant adaptation to waterlogging stress. ET improved photosynthesis in mustard and wheat plants subjected to salinity stress (*Nazar et al., 2014*). According to *Wang et al. (2020a)*, ET increased seed germination in alfalfa under salt stress and ameliorated the effects of salt stress on seedlings. ET's role in cell senescence and plant maturation best describes it as an 'aging' hormone (*Schaller, 2012*).

## Brassinolide (BL)

Brassinolide is a plant growth-promoting steroid and the most significant naturally occurring brassinosteroid due to its effective biological functions and widespread distribution (*Li et al., 2020*). In tomato, exogenous BL significantly improved vegetative growth by increasing root/shoot length and biomass, shoot architecture, and total chlorophyll (a & b), carotenoids, carbohydrate as well as mineral contents (*Nafie & El-Khallal, 2000*). In potatoes, the application of 24-epibrassinolide (EBL) prolonged dormancy, increased ET production and improved ABA content in buds (*Korableva et al., 2002*). Also, exogenous BL improved nitrate reductase activity, absorption of nitrate fertilizer, plant height, biomass accumulation, and resistance to adverse environmental conditions in diverse plant species including chickpea, soybean, wheat, lentil, maize, and rice (*Ali, Hayat & Ahmad, 2007*; *Anjum et al., 2011*; *Hayat & Ahmad, 2003*).

## Salicylic acid (SA)

Salicylic acid (2-hydroxybenzoic acid) is a phenolic compound synthesized naturally in plants (cytoplasm and chloroplast), most of which are methylated and/or glucosylated (*Janda & Ruelland, 2015*). SA is known for its cardinal functions in regulating various biochemical and physiological processes in plants including photosynthesis, nutrient transport, absorption of ion, stomatal closure, gaseous exchange, protein synthesis, flowering, seed germination, senescence (*Davies, 2010*), and response to environmental stresses (*Simaei, Khavari-Nejad & Bernard, 2012*). For example, the application of SA increased the number of branches, plant height, and overall yield in cotton (*Al-Rawi, Al-Ani & Al-Saad, 2014*). In thyme (*Thymus vulgaris* L), the rate of photosynthesis, dry weight, and tolerance to salt stress increased with a similar observation in rosemary (*Rosmarinus officinalis* L; *Najafian, Khoshkhui & Tavallali, 2009*). Under various abiotic

stress conditions, the exogenous application of SA at lower concentrations (0.05–0.5 mM) significantly increased seed germination and seedling establishment in *Arabidopsis* and to an extent alleviated the effects of heat and oxidative stress on seed germination (*Alonso-Ramírez et al., 2009*).

SA facilitates plant's defense response against microbial pathogens by promoting the production of pathogenesis-related (PR) proteins and upregulating its level in response to pathogen attack, hence, associated with plant immunity (*Wildermuth et al., 2001*). For instance, the exogenous application of SA actuated the production of PR proteins and induced tolerance to tobacco mosaic virus (TMV) in tobacco plants. This became evident as SA accumulated in the TMV-infected tobacco plants with a similar observation in *Pseudomonas syringae* pv. *tomato* (*Pst*) infected cucumber plants (*Raskin, 1992*). All the above effects were concentration-dependent, in that different application levels stimulated varied results.

## Jasmonic acid (JA)

Jasmonic acid (3-oxo-2-2′-*cis*-pentenyl-cyclopentane-1-acetic acid), is a derivative of beta-linolenic acid, an essential component of chloroplast membranes belonging to the oxylipins (*Mosblech, Feussner & Heilmann, 2009*). JA is associated with key biological and physiological processes such as growth and development by mediating the transport of photosynthates (*Kaplan, 2012*), uptake of nutrients (nitrogen and phosphorus), inducing stomatal opening, and stress response (*Wasternack & Hause, 2013*). JA doubles as a promoter and inhibitor of growth. For instance, exogenous JA suppressed leaf expansion and adventitious root formation in *Arabidopsis* (*Gutierrez et al., 2012*), decreased coleoptile growth and plant height in rice, and inhibited ear shoot growth in maize (*Riemann et al., 2013*; *Yan et al., 2012*). However, JA was found to positively regulate lateral root formation in *Arabidopsis* (*Cai et al., 2014*) and fiber elongation in cotton (*Hao et al., 2012*). In Safflower, the foliar spray of JA increased the maximum quantum yield of photosystem II (*Fv/Fm*), relative water content, chlorophyll content, biomass accumulation, and grain yield (*Ghassemi-Golezani & Hosseinzadeh-Mahootchi, 2015*). Furthermore, the exogenous application of JA induced chlorophyllase activity, leaf senescence, and microtubule degradation of soybean plants (*Hassanein et al., 2009*).

The signaling pathway of JA serves as an effective mediator of environmental stress response, mostly oxidative damage and pathogenic infections, and thus, induces the expression of several target genes suited for this function (*Gupta et al., 2017*). For instance, *Faghih, Ghobadi & Zarei (2017)* revealed that the exogenous application of JA to strawberry plants subjected to salt stress decreased lipid peroxidation by improving the antioxidant enzyme activity. Also, JA increased the metabolism of glutathione and ascorbate in the leaf tissues of crested wheatgrass subjected to water stress, thereby increasing tolerance to water stress (*Shan & Liang, 2010*). In *Arabidopsis*, priming with JA increased tolerance to drought by inducing the expression of reactive genes associated with drought stress (*Liu & Avramova, 2016*). JA stimulates the activities of other plant hormones like ABA, ET, and polyamines (*Onkokesung et al., 2012*). For instance, JA application delayed ABA-mediated

inhibition of seed germination in *Arabidopsis* (*Ellis & Turner, 2002*) and elevated the content of spermidine in barley (*Bandurska, Stroiński & Kubiś, 2003*).

## Amine compound

Amine compounds are generally categorized into primary, secondary, and tertiary amines depending on the number of amine groups (one, two, and three respectively). In plant growth and development, the majority of research has focused on polyamines and substituted tertiary amines.

Polyamines (PAs) are low molecular weight aliphatic nitrogen compounds comprising of two or more amino groups with strong physiological activity (*Vuosku et al., 2018*). PAs are classified as novel PGRs usually distributed in virtually all living organisms. The main polyamines present in plants include putrescine (Put), spermidine (Spd), thermospermine (Tspm), and spermine (Spm) and are implicated to regulate several biological processes including embryogenesis, fruit and flower development, and senescence (*Mustafavi et al., 2018*). The distribution of individual polyamines is tissue/organ-specific with some localized in different cells of the same tissue, attributed to their distinctive functions. For instance, Put was revealed as the most prevalent PA in leaves and its levels were three times higher compared to Spd and Spm while other organs had Spd as the most dominant PA (*Takahashi et al., 2018*). In carrot, Put and Spm were found to accumulate in the cytoplasm and cell wall respectively (*Cai, Zhang & Guo, 2006*).

PAs play pivotal roles in regulating plant defense response to diverse stress signals. For instance, the exogenous application of PA has been reported to increase stress tolerance in plants leading to improved yield (*Liu et al., 2007*). According to *Liu et al. (2007)*, the genetic modification of plants with polyamine biosynthesis genes enhanced abiotic stress tolerance in resulting transgenic plants. Put applied exogenously to chickpea, alfalfa, and welsh onion enhanced tolerance to cold, drought, and waterlogging stress respectively (*Nayyar, 2005*; *Yiu et al., 2009*; *Zeid & Shedeed, 2006*). The exogenous application of spermine and spermidine improved growth and yield in *Arabidopsis* and tomato by enhancing tolerance to drought and heat stress respectively (*Murkowski, 2001*; *Yamaguchi et al., 2007*). PAs were identified to stimulate embryogenesis in Asian ginseng (*Panax ginseng*; (*Kevers et al., 2000*) and promote shoot regeneration in Korean and Chinese radish (*Curtis, Nam & Sakamoto, 2004*; *Pua et al., 1996*) and chili pepper (*Capsicum frutescens*; (*Kumar et al., 2007*).

Substituted tertiary amines (STAs) also increase crop productivity by increasing root growth, photosynthetic efficiency, and overall plant vigor (*Qi et al., 2013*). The most commonly used STAs in crop production include but not limited to 2-(3, 4-dichlorophenoxy) trimethylamine (DCPTA), diethyl-2-(4-methylbenzoxy) ethylamine (MBTA), diethyl aminoethyl hexanoate (DTA-6), and 2-(N-methylbenzylaminoethyl)-3-methylbutanoate (BMVE). Studies have shown a positive yield effect of DCPTA application in tomatoes (*Keithly, Yokoyama & Gausman, 1990*) and a promotive effect on seedling growth and development in radish (*Keithly, Yokoyama & Gausman, 1992*). In another study, seed treatment with 30 μM DCPTA resulted in a significantly enhanced root and hypocotyl elongation along with seedling dry weight of radish (*Keithly, Yokoyama & Gausman, 1992*). *Van Pelt & Popham (2007)* revealed that the foliar application of DCPTA

and MBTA during the two true leaf stage of pepper seedlings did not only increase yield but also significantly improved fruit pigment content. *Wang et al. (2016)* recently showed that DCPTA application increased plant growth in maize but the resulting stem elongation increased lodging and subsequently decreased yield. However, the combined effect of DCPTA and CCC sprayed at the seedling stage ameliorated the problem of lodging by reducing stem elongation leading to increased yield. This is because CCC acts antagonistically to gibberellic acid to inhibit stem elongation. Seed treatment with BMVE reportedly increased plant growth and yield in multiple crop species (*Yokoyama et al., 2015*).

According to *Qi et al. (2013)*, the foliar treatment of DTA-6 on the canopy of maize and soybean at the third trifoliate (V3) stage increased plant height, root length, leaf area, dry matter accumulation, root to shoot ratio, along with improved Rubisco and phosphoenolpyruvate carboxylase (PEPCase) activity, chlorophyll content, and photosynthetic rate with similar observations in radish, spinach, and tomato (*Keithly, Yokoyama & Gausman, 1992*). Furthermore, the application of DTA-6 increased $CO_2$ assimilation rate in maize and soybean seedlings as the plants exhibited higher *Fv/Fm* values than the control and ultimately improved the photosynthetic apparatus.

## REGULATION EFFECTS OF DIFFERENT PGRS ON SOYBEAN

Application of PGRs control plant development by regulating the endogenous hormone biosynthesis and catabolic system to facilitate the desired growth (*Zeng et al., 2012*). PGRs have numerous advantages over traditional methods of crop production due to their effectiveness at low concentrations, wide range of applications, low toxicity, ability to regulate plant morphology and physiology, and their influence on many crop species (*Saini et al., 2013a*). They affect plant growth from embryo development to completion of life cycle and death (*Li et al., 2010*). The physiological effect of PGRs depends on the time and method of application, concentration, frequency of application, weather conditions, and state of plant (*Dadnia, 2011*). Several studies suggest the positive effects of growth regulators on the morphology, physiology, yield, and yield quality of crops, especially soybean. The various actions of PGRs in regulating the different aspects of soybean growth and development are summarized in Fig. 1.

### Effects on soybean morphology

Plant morphology generally consists of the root and shoot systems. The root system encompasses various types of roots, dynamic in morphology and functions. The vital features that describe root morphology include but not limited to root length and diameter, root surface area (RSA), and root volume (*Wijewardana, Reddy & Bellaloui, 2019*). Other features like lateral roots, forks, crossings, and tips regulate the root architecture which is also controlled by the distribution and spatial arrangement of the roots in the soil by positioning its foraging activity to regulate water and nutrient absorption. The shoot system also encompasses plant parts above the soil, usually the leaves, stem, and branches generally

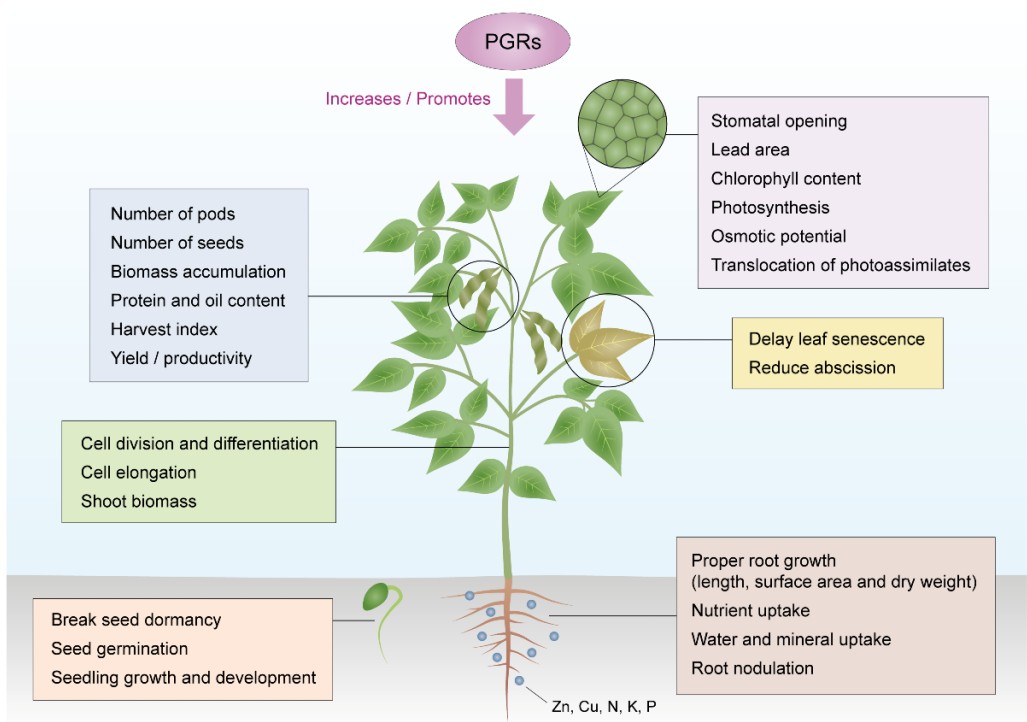

**Figure 1 Action of PGRs in regulating growth and development of the soybean plant.** Source modified from *Nadeem et al. (2019)*.

termed as the shoot. This generally determines the plant's height. These two major systems are essential to plants and are highly influenced by PGRs as detailed below.

### Root

Botanically, the otanically, the root is an underground organ and functions primarily in absorbing soil minerals and moisture and as well offers firm anchorage. Roots, as the central boundary between plants and the surrounding environment, exert key functions in the growth and development of plants (*Marschner, 2011*). Root growth and development directly affect the growth of the soybean plant. With PGRs affecting growth, understanding their influence on the roots is very important.

In soybean growth, most root traits arise from the features of the soybean plant rather than the soil unlike in most crops where the root length and weight are dependent on the oxygen content of the soil which is ascertained by the soil porosity (*Yang, Wu & Zhang, 2001*). Soybean roots under various soil parameters such as moisture, temperature, voids can cause variations in root morphology in addition to other environmental factors (*Jian, Xiaobing & Guanghua, 2002*). Conditions such as waterlogging stress and drought highly influence root growth. Waterlogging stress inhibits root growth by decreasing the size of the root whiles drought influences the root architecture through longer lateral root and root hairs development for better water absorption (*Kim et al., 2018*). Waterlogging fills soil pores which limits gaseous exchange, thus triggering adventitious root formation

(*Steffens, Wang & Sauter, 2006*). PGRs are positive promoters of growth, with ET regulating adventitious root formation. The availability of ET and GA$_3$ triggered a substantial increase in the development and penetration of adventitious roots of several plants (*Steffens, Wang & Sauter, 2006*).

In soybean, endogenous GA increased with the application of ET coupled with increased plant height and adventitious root formation, proposing the ability of ET to improve GA accumulation and hence, improve root growth (*Kim et al., 2018*). The application of 2-chloroethylphosphonic acid (ethephon, ETP) increased the RSA of the soybean plant, providing an increased surface area for water absorption (*Kim et al., 2018*). ET and GA control the formation, number, and length of adventitious roots synergistically, hence, exogenous ET application boosts GA accumulation in soybean plants which enhance the RSA and improves root growth. Exogenous application of IBA to soybean hypocotyl induced root development by significantly increasing the number of lateral roots (*Chao et al., 2001*).

### Shoot

The shoot of plants encompasses its stem and leaves. The leaves function to effectively capture photosynthetically active radiation (PAR). Leaf development is thereby well-thought-out as the principal phenomena of growth and shoot morphogenesis during canopy developmental stages in plants. The application of PGRs is often used as a measure to ensure the reduction of lodging incidence by mimicking or altering the production of hormones essential in improving stem structure and increasing yields (*Berry & Spink, 2012*).

In soybean, the application of ETP showed greater shoot growth resulting in increased plant height (*Kim et al., 2018*). The increase may be related to the elongation of internodes and heightened cell division and expansion. Growth regulator treatments increased the leaf area per plant and leaf area index (*Manu, Halagalimath & Chandranath, 2020*). For instance, cycocel treatments enhanced a prolonged assimilation surface area which delayed leaf senescence, hence, retaining more leaves per plant. Also, cycocel reduced chlorophyll degradation, protease activity, and stimulated the synthesis of soluble proteins and enzymes (*Khalilzadeh, Seyed Sharifi & Jalilian, 2018*).

Cytokinin is known to induce shoot regeneration and elongation. Its application was observed to control shoot branching and stimulate the growth of axillary bud (*Shimizu-Sato, Tanaka & Mori, 2009*). Exogenous application of cytokinin alleviated lateral branch inhibition in soybean plants subjected to high-aluminum toxicity, thus, promoting shoot growth (*Pan, Hopkins & Jackson, 1988*). Although cytokinin has been observed to increase shoot development in several plants, very little information is available on soybean shoot development, hence, presenting a gap for future research. Soybean cv. BS-3 sprayed with 100 ppm of IAA at three different times increased plant height, number of flowers, pod number, percent of fruit set, seed number per plant, seed yield per plant, and total seed yield (t/ha) (*Sarkar, Haque & Abdul Karim, 2002*). Application of 100 ppm NAA at the flowering stage increased the number of branches per plant, plant height, leaf number per plant, average pod weight, leaf area, dry matter, and seed yield (*Deotale et al., 1998*).

### Root nodules

Root nodules are the nitrogen-fixing cell protrusions that contain Gram-negative bacteria found on the roots of leguminous plants. These protuberances are formed through a process known as nodulation. Nodulation provides atmospheric nitrogen ($N_2$) needed for the synthesis of nitrogen-containing compounds like nucleic acids and proteins necessary for enhancing plant growth physiognomies, crop yield, and conserve soil fertility (*Mangena, 2018*). According to *Ohyama et al. (2011)*, one element positively influencing soybean productivity is biological nitrogen fixation. However, it is established through research that the root nodules of legumes including soybean improve soil fertility through atmospheric nitrogen fixation and subsequently, increasing yield. Several studies have revealed that exogenous PGRs promote nodule formation (*Mishra et al., 1999*). By import, plant growth regulators could improve soybean yield as it promotes nodulation (*Sudadi & Suryono, 2015*).

Nodule formation in soybean is extremely sensitive to exogenous plant growth regulators such that they intricately regulate nodule development and affect nitrogen fixation. For instance, the addition of L-tryptophan and indole acetic acid to sterile sand media of neutral pH increased the number of nodules (*Sudadi, 2012*). Likewise, the addition of epibrassinolide to growth media improved nodulation in soybean by increasing the number of nodules per plant. *Sudadi & Suryono (2015)* reported that L-tryptophan (0.001 mg $L^{-1}$, 0.1 mg $L^{-1}$, 1.0 mg $L^{-1}$) increased the number of root nodules per plant. Similarly, *Basuchaudhuri (2016)* reported that applying tryptophan and IAA induced root nodulation in soybean. In another study, the exogenous application of 6-Benzylaminopurine (BAP), indole acetic acid, salicylic acid, gibberellic acid, and jasmonic acid increased nodule number per plant compared to untreated plants, though high concentrations of $GA_3$ (1 $\mu$M) and JA (100 $\mu$M) significantly decreased nodule number by about 70% and 95% respectively. The results revealed that the effect of these growth regulators is concentration-dependent, however, 10 nM IAA, 50 nM BAP, 10 $\mu$M JA, 10 nM, and 100 nM $GA_3$ and 100 $\mu$M to 1 mM SA had the best effect and is considered ideal for high nodule numbers in soybean (*Roy Choudhury, Johns & Pandey 2019*). According to *Mens et al. (2018)*, the exogenous application of cytokinins (BAP, $N^6$-($\Delta^2$-isopentenyl)-adenine and *trans*-zeatin) at low concentrations promoted nodule development while high concentrations decreased nodulation in soybean. Seed priming with low cytokinin concentrations ($10^{-9}$ mol/L) promoted nodulation by increasing the total nodule area, thereby improving biological nitrogen fixation under controlled environments (*Kempster et al., 2021*). By this, it can be suggested that plant growth regulators intricately regulate nodulation and thus, effective nodulation requires strict regulation of the concentration of growth regulators applied.

## Effects on soybean physiology

Generally, plant physiology encompasses all processes and mechanisms that make a plant functional such as photosynthesis, water utilization, and environmental physiology among others (*Hopkins & Huner, 2008*). These mechanisms control the growth and development of both shoots and roots, resistance to environmental perturbations, and the interaction of factors associated with growth like nutrition, hormones, temperature, carbon, and energy

metabolism. In soybean, all the above mechanisms have been researched with this review focusing on how PGRs boost the utilization of hormones, fertilizer, water, enzymes, and photosynthesis.

### Hormone

Plants produce chemicals that regulate processes of growth and development like reproduction, continuous survival, and senescence (*Mustafa et al., 2016*). These endogenous chemical substances known as phytohormones (plant hormones) are a product of plant secondary metabolism, acting as chemical messengers that coordinate a host of signaling pathways and regulate varied cellular responses that facilitate plants' adaptation to abiotic stress (*Kazan, 2015*; *Fahad et al., 2015*). Application before/during stress exposure may modulate the endogenous hormone levels to activate plants' response to stress (*Kaur, Gupta & Kaur, 1998b*). Fluctuations in levels of the major plant hormones (auxin, abscisic acid, cytokinin, ethylene, and gibberellins) influence all aspects of plant growth (*Zhao et al., 2012*). These hormones intricately regulate plant growth by either promoting or inhibiting growth from germination to reproductive growth and flowering, hence classified as growth promoters (gibberellins, cytokinin, and auxin) and growth retardants (abscisic acid and ethylene; *Nadeem et al., 2019*). Under normal growth and environmental conditions, these hormones are naturally synthesized in certain plant parts but their production minimizes when conditions are unfavorable to promote optimal growth and productivity (*Zahir, Asghar & Arshad, 2001*). Synthetic compounds that mimic the activities of plant hormones otherwise known as PGRs are exogenously applied to alter the endogenous hormone levels in plants and to augment growth and yield (*Mustafa et al., 2016*).

Plant growth regulators increase endogenous hormones in plants. In soybean, exogenously applied methyl jasmonate (MeJA) increased the abscisic acid content in plants subjected to salt stress, thus, accelerating tolerance to salt stress (*Yoon et al., 2009*). A report by *Hamayun et al. (2015)* revealed that exogenous application of kinetin (Kn) significantly increased the endogenous bioactive gibberellins ($GA_1$ and $GA_4$), jasmonic acid, and free salicylic acid contents in soybean plants subjected to salt stress. This was evident as Kn up-regulated the biosynthesis pathways of gibberellins, jasmonic acid, and free salicylic acid unlike the control. *Qi et al. (2013)* reported that the foliar application of DTA-6 on soybean leaves increased the endogenous hormone (Zeatin Riboside, gibberellins, and indole acetic acid) levels. This increased PEPCase and Rubisco (key photosynthetic enzymes) activity, photosynthetic rate, high biomass accumulation, and consequently increased productivity. In conclusion, exogenous PGRs regulate the endogenous hormone levels to increase plant metabolic processes associated with growth and productivity.

### Enzyme

Enzymes are mostly utilized in both plants and animals as an effective defense mechanism for eliminating reactive oxygen species (ROS). ROS induces cellular oxidative damage which impairs cell components with a concomitant effect on cell function leading to death (*Foyer & Noctor, 2011*). As reviewed by *Hasanuzzaman et al. (2020)*, various enzymes like peroxidase (POD), catalase (CAT), *superoxide dismutase* (SOD), and glutathione-s-transferase (GST)
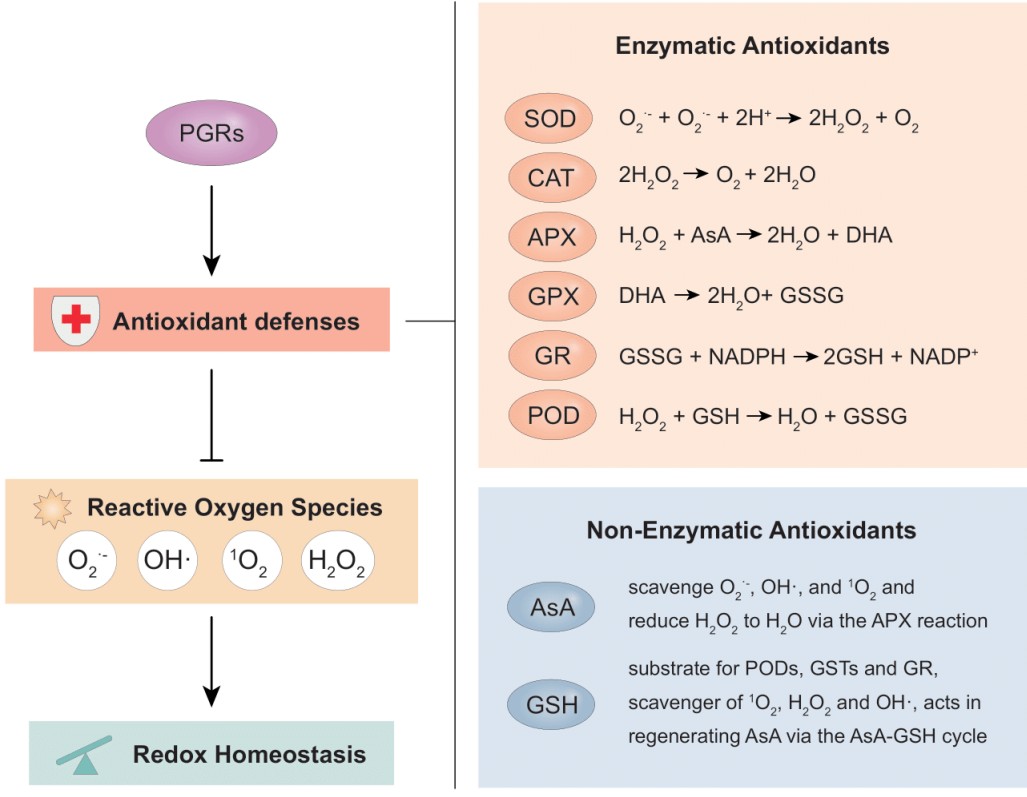

**Figure 2** **The main antioxidant defense system and the various reactions involved in scavenging ROS from cells.** CAT, catalase; APX, ascorbate peroxidase; POD, peroxidase activity; SOD, superoxide dismutase; GR, glutathione reductase; GSH, glutathione; NADPH, nicotinamide adenine dinucleotide phosphate; GSSG, glutathione disulfide; GPX, glutathione peroxidase; GST, glutathione S-transferase; AS-C/AsA, ascorbate; DHA, dehydroascorbate; $^{1}O_2$, singlet oxygen; $H_2O$, water; $O_2$, molecular oxygen; $OH^{\bullet}$, hydroxyl radical; $H_2O_2$, hydrogen peroxide; and $O_2^{\bullet-}$, superoxide radical. In the long process of natural evolution, plants themselves have established a set of extremely complex antioxidant defense mechanisms to avoid or alleviate oxidative damage. As the main defense system against ROS *in vivo*, the antioxidant defense consisting of a systematic network of both enzymatic and non-enzymatic antioxidants work coordinately to scavenge free radicals from cells. SOD enzyme catalyzes the decomposition of $O_2^{\bullet-}$ into $H_2O_2$ which may react with membrane lipids to undergo a chain of reactions and further be converted to $H_2O$ and $O_2$ by various antioxidants such as CAT, GSH, GSSG, GPX, AsA, DHA, and GR. The $OH^{\bullet}$ generated by $H_2O_2$ is further scavenged by GSTs. The non-enzymatic antioxidant system, on the other hand, does not directly protect against ROS but works in synergy with the endogenous enzymatic antioxidant system to scavenge free radicals by augmenting their functions.

in addition to non-enzymatic antioxidants such as reduced ascorbate (AsA) and glutathione (GSH) cooperatively reduce oxidative damage under stress conditions (Fig. 2).

SOD degrades excess ROS by transforming $O_2^{\bullet-}$ to $O_2$ and $H_2O_2$ (*Bresciani, Da Cruz & González-Gallego, 2015*) whiles CAT catalyzes the conversion of $H_2O_2$ to $H_2O$ (*Furukawa et al., 2017*). Both antioxidant enzymes are essential and considered as a preliminary defense mechanism against oxygen toxicity. Ascorbate peroxidase (APX) is a widely distributed antioxidant in plant cells with its isoforms having a higher affinity for $H_2O_2$ than CAT, rendering APXs as an efficient scavenger of $H_2O_2$ under stressful conditions

(*Sharma, Jha & Dubey, 2019*). GSH is an essential redox molecule found in all living cells and forthrightly associated with antioxidant defense and maintenance of intracellular redox homeostasis by stimulating ascorbic acid synthesis, through the Ascorbate-Glutathione cycle (*Wang, Yang & Yi, 2012*). GSH also plays a direct role as a free radical scavenger due to its ability to react with $H_2O_2$, $OH^\bullet$, and $O_2^{\bullet-}$. In soybean, POD, APX, polyphenol oxidase (PPO), and CAT activities were up-regulated under salinity stress with a significant increase in MDA and hydrogen peroxide. Also, applying PGRs stimulated the expression of these enzymes hence, influencing stress response. PGRs (salicylic acid and nitric oxide) inhibited the production and accumulation of ROS ($H_2O_2$, $O_2^{\bullet-}$, $OH^\bullet$), decreased lipid peroxidation and increased proline accumulation under salt stress (*Tan et al., 2008*). This significantly decreased MDA and increased the total antioxidant activity (by scavenging ROS which impeded its harmful effects), thereby increasing tolerance to salt stress. In soybean, exogenously applying SA, sodium nitroprusside (SNP), and their combination (SA+SNP) regulated the proline levels and reduced salt-induced oxidative damage marked by a reduction in the MDA content and subsequently, decreased lipid peroxidation and enhanced catalase activity (*Simaei et al., 2011*). Applying ETP alleviated water stress in soybean plants by up-regulating GST3 and GST8 (*Kim et al., 2018*). ETP significantly enhanced the expression of GST3 and GST8 at the biochemical and transcriptional level. Hence, an increase in GST3 and GST8 levels heightened GSH activity, reduced reactive oxygen species, alleviated cell damage in photosynthetic apparatus, and enhanced phenotype (*Kim et al., 2018*).

Foliar application of NAA improved nitrate reductase activity in soybean (*Senthil, Pathmanaban & Srinivasan, 2003*). Also, the application of SOD simulation material ($SOD_M$) and DTA-6 increased SOD, CAT, and POD activities in soybean leaves and lowered MDA, resulting in enhanced seed yield and delayed leaf senescence (*Zheng et al., 2008*). DTA-6 and uniconazole (S3307) increased the POD activity in pods, reduced vital abscission enzyme activity, MDA activities, and lipid peroxidation, thereby enhancing the accumulation and transportation of assimilates as well as antioxidant capacity (*Sun et al., 2016*). In addition, exogenously applied SNP induced the antioxidant machinery in soybean plants by enhancing various antioxidant enzyme (CAT, SOD, POD, and APX) activities (*Jabeen et al., 2021*).

Brassinosteroids and JA are well known for their antioxidant capacity and their major involvement in regulating the AsA-GSH cycle in plants. These PGRs boost the plant defense system by increasing antioxidant system capacity, enzymatic activity, and upregulating the expression of antioxidant genes (APX, SOD, POD, CAT, and glutathione reductase (GR); *Bali et al., 2018*). In soybean, seed priming with EBL increased the antioxidant enzyme (CAT, APX, SOD, and GR) activity and the contents of non-enzymatic antioxidants such as AsA, GSH, and glutathione disulfide (GSSG) under salt stress, which improved salt tolerance, biomass accumulation, and subsequently yield (*Soliman et al., 2020*). Similar results were obtained with exogenous EBL application in salt-stressed soybean plants as reported by *Alam et al. (2019)*. Furthermore, exogenous 28-homobrassinolide enhanced the AsA-GSH cycle by activating the antioxidant system, marked by improved activities of APX, SOD, CAT, monodehydroascorbate reductase (MDHAR), dehydroascorbate

reductase (DHAR), and GR), leading to efficient ROS scavenging in soybean seedlings (*Hasan et al., 2020*). According to *Mir et al. (2018),* the exogenous application of JA enhanced the contents of non-enzymatic antioxidants and antioxidant enzyme activity, thereby boosting redox homeostasis and antioxidant capacity in Nickel-stressed soybean plants. Similar results were also reported by *Sirhindi et al. (2016)*. In another study, MeJA treatment increased CAT, SOD, and APX activity and the level of dehydroascorbic acid (DHA) in soybean leaves under cadmium stress, thereby ameliorating cadmium toxicity (*Keramat, Kalantari & Arvin, 2009*). Eventually, all these resulted in heightened growth and productivity of the soybean plant.

### Photosynthesis

Photosynthesis is a vital phenomenon in crop production as it provides the energy needed for plant growth and development, thus, regarded as the basis of growth in plants. *Murata (1981)* reported a relatively high positive correlation between the optimal growth rate and leaf photosynthesis of many crops. However, the growth rate of plants may not directly reflect their photosynthetic rate. Leaf photosynthesis is a vital component of crop biomass production, thus, an essential determining factor of grain yield through canopy photosynthesis. For instance, *Connor, Loomis & Cassman (2011)* revealed that photosynthesis produces simple carbohydrates from which more complex carbohydrates (proteins and lipids) are obtained to form the dry matter of plants. In soybean, high seed yield results from enhanced canopy photosynthesis after seed germination. It is hypothesized that a decline in canopy photosynthesis significantly reduced soybean productivity, attributed to decreased pod setting (*Jiang & Egli, 1993*). Furthermore, the improved production of assimilates at the fruit development stage in soybean may increase yield. Based on the above results, it can be deduced that enhanced photosynthesis may improve growth, biomass production, and subsequently increase grain yield in soybean. Therefore, improving photosynthesis in soybean is essential as it may increase productivity without altering other genetic factors (*Long et al., 2006*).

Plant growth regulators enhance chlorophyll content, improve the photosynthetic ability, and increase translocation of assimilates from source to sink , thereby increasing biomass production and ultimately yield. *Shao et al. (2014)* reported a positive correlation between the rate of photosynthesis and chlorophyll content, indicating that chlorophyll content highly influences photosynthesis in plants. However, this relationship is limited in chlorophyll range, in that increased chlorophyll production may not necessarily increase photosynthetic rate. Generally, cycocel is reported to increase the translocation of photosynthates (*Grewal et al., 1993*). Foliar spray of cycocel (500 ppm) on soybean leaves at flower initiation (R1), pod initiation (R3), and flower + pod initiation (R1+R3) growth stages increased the chlorophyll content (*Devi et al., 2011*). The marked increment in chlorophyll content improved photosynthesis and greatly enhanced the translocation of photoassimilates to sink organs (seeds) leading to increased biomass accumulation. Similarly, the foliar spray of NAA (40 ppm), $SOD_M$, DTA-6, and ABA (300 mg/l) on soybean plants increased the rate of photosynthesis and chlorophyll content of soybean plants (*Qi et al., 2013*; *Travaglia, Reinoso & Bottini, 2009*; *Zhao et al., 2008*). The main

factors for the enhanced photosynthesis in the DTA-6 treated plants were the increased activities of key photosynthetic enzymes (PEPCase and Rubisco), improved electron transport activity which elevated the rate of $CO_2$ assimilation, eventually improving the photosynthetic apparatus.

Also, DTA-6 and uniconazole significantly increased leaf photosynthetic ability. These growth regulators maintained high photosynthetic activity, increased photosynthetic rate, and subsequently biomass accumulation due to enhanced translocation of photosynthates (sucrose and starch) to different organs resulting in improved yield (*Liu et al., 2019*). Exogenously applied SA maintained the stability of PSII, increased chlorophyll a/b binding proteins (Chla/bBP), Rubisco activase, Rubisco subunits, oxygen-evolving enhancer protein 1 and 2 (OEE1 and OEE2, respectively), ferredoxin NADP reductase (FNR), and photosynthesis-related proteins in soybean. This improved stomatal conductance, photosynthetic rate, and water use efficiency (WUE), hence, increasing photosynthesis. Also, antioxidant enzyme activity increased with SA treatment (*Sharma et al., 2018*). These findings suggest that plant growth regulators enhance photosynthesis in soybean by increasing the chlorophyll content, distribution of photosynthates, and biomass accumulation which eventually improves growth and yield.

As recently reviewed by *Müller & Munné-Bosch (2021)*, hormonal crosstalk positively regulates photosynthesis in plants. For instance, the exogenous application of IAA caused a marked increase in endogenous JA and ABA levels leading to significantly higher chlorophyll contents, thus, elevating the rate of photosynthesis in clover (*Zhang et al., 2020*). Furthermore, exogenous BAP increased endogenous zeatin levels which enhanced the $F_v/F_m$, ΦPSII (relative quantum yield of PSII), and electron transport activity in wheat (*Yang et al., 2018*). In soybean, MeJA treatments resulted in elevated levels of ABA which increased the photosynthetic rate (*Yoon et al., 2009*). In addition, the foliar spray of DTA-6 increased endogenous Zeatin Riboside, gibberellins, and IAA levels which increased PEPCase & Rubisco levels and the rate of photosynthesis in soybean. This enhanced biomass accumulation, thereby improving productivity (*Qi et al., 2013*). Though there is extensive research to support the positive regulation of hormonal crosstalk on photosynthesis in several plants, that of soybean is limited, thus, presenting a gap for future research.

### Water utilization

Water utilization is very essential in the growth of plants. Every level of development requires a certain amount of water which is mostly dependent on the season, place and time of planting, and cultivars involved. The sufficient quantity of water available at these times of growth and development is essential for the diverse parameters of growth such as seed weight and size among others.

Generally, water use efficiency (WUE) defines the quantity of carbon assimilated as biomass produced per unit of water used by the plant. Hence, WUE is expressed as the relationship between yield/biomass to the soil water evaporation component and transpiration component which together denotes Evapotranspiration (ET) or the overall water available to the crop, comprising precipitation and the quantity of water available through irrigation *i.e.,* Yield = ET*WUE where ET = $T_{soil}$ + $T_{crop}$ (*Kuglitsch et al., 2008*).

Understanding plant water use is further detailed down by analyzing the water use at the leaf and canopy levels. The leaf level WUE is expressed as the net photosynthetic rate ($A_n$) divided by the transpiration rate ($E$) (*Hatfield & Dold, 2019*). In soybean, analysis of WUE at leaf and field scales revealed that a 1% increase in leaf-scale WUE showed ∼10% increment in field-scale explaining the 90% yearly variability in field-scale (*Gorthi, 2017*). WUE can hence be utilized as secondary criteria for selecting seed yield genotypes.

PGRs, being popular for their influence on various morphological and physiological parameters of plants, have limited information regarding water utilization in soybeans. According to *Kamal et al. (1998)*, the foliar application of 2-ppm S-abscisic acid on soybean at the reproductive stage increased seed yield, pod number, and seed number under conditions of optimal or moderate water stress whereas severe conditions reduced seed yield. The WUE of soybean was enhanced after the application. GA$_3$ application during flowering and pod development, significantly increased rainwater use efficiency and productivity (*Giri et al., 2018*). This then proposes that GA$_3$ application enhanced the effective use of water by the soybean plant which was reflected in the increment in its yield.

*Fertilizer utilization and nitrogen fixation*

Nitrogen is a key nutrient for plant growth. Generally, legumes can fix nitrogen in soils which are further utilized by the plant for growth (*Wang et al., 2020b*). Di-nitrogen ($N_2$) fixation by leguminous crops has attracted much interest in recent times due to the nitrogen nutrition it provides in a highly sustainable and economically competitive way hence, participating in environmentally sound agricultural production as well as high-quality crop products (*Vollmann et al., 2011*).

In soybean, an increase in growth and yield have been reported to be dependent on large inputs of nitrogen with the pod initiation (R3) to full seed (R6) growth periods, requiring higher N contents (*Oplinger, 1991*). Despite this high requirement of nitrogen, only 25–60% of this nitrogen is naturally available with the rest compensated for by the application of nitrogenous fertilizers to the soil. Although there have been several conflicting reports on the benefits of applying nitrogenous fertilizers for soybean yield, it has been firmly agreed that soybean plants act as sinks for soil nitrogen and utilize it effectively (*Varvel & Peterson, 1992*). The effective utilization of this soil nitrogen during the early pod filling stage increased yield (*Oplinger, 1991*). However, increasing soil nitrogen at planting reduces soybean response to N fertilizer (*Stone, Whitney & Anderson, 1985*).

Despite all the benefits stated by several researchers about exogenous nitrogen application, it can be observed that the benefits were directly related to the time of application. For instance, the application of N fertilizer at planting reduced nodule formation, increased grain yield at flowering and early pod fill growth stages (*Oplinger, 1991*). Also, starter N application at the beginning flowering/flower initiation (R1) stage did not influence protein/oil concentrations or increase yield but increased dry matter content and plant N content (*Wood, Torbert & Weaver, 1993*). At the beginning seed (R5) stage, N increased grain yield, dry matter accumulation, seed protein concentration but reduced seed oil concentration. In all, with N fertilizer application improving soybean growth during the early season, it can hence be proposed that application at this stage is

beneficial to the soybean plant and its nitrogen content, however, for effective utilization of the applied fertilizer, the R5 stage may be the most reliable. The above results, however, may not be independent of other factors. Per speculations, environmental factors may limit soybean growth by restricting N fixation, which may result in a positive response to N fertilizer. Hence, factors such as nitrogen concentration, cultivar, and environmental parameters may be responsible for several conflicting reports in terms of fertilizer utilization in soybean plants.

Despite all the above reports, one major limitation is the fact that the above researches were conducted one to three decades ago. Current information on fertilizer utilization is however very limited with none on how PGRs regulate fertilizer utilization. However, from knowing the various impacts and limitations of PGRs and fertilizers, the following hypothesis can be made. With the majority of PGRs involved in improving growth and yield, their application together with N fertilizers might enhance growth and yield in double folds than the individual increases and probably reduce the negative effects such as low nodule formation. A study by *Devi et al. (2011)* indicated that applying the required amount of NPK fertilizer at planting before applying various PGRs (50 ppm Salicylic acid, 200 ppm Ethrel, and Cycocel) at three different stages of development (R1, R2, and R1+R2) increased growth, yield and yield components, photosynthetic pigments, and protein and oil contents. Although it was not categorically stated that the result was due to the fertilizer application, there is a possibility of the fertilizer playing a role and revealing the chance of our hypothesis being true. This hypothesis, however, needs to be thoroughly investigated to reach a conclusive finding.

## Effects on soybean yield, yield components, and quality

In soybeans, the yield production process highlights those processes and features that play vital roles in defining yield. These features, mostly termed as the yield component includes seed biomass, pod number, seed number, and seed size, among others. The quality of the soybean also defines a successful harvest and yield.

### Yield and yield components

Major evidence of the extent of growth in plants is their yield. The yield is hence known as the productivity or output of the plant. Several traits of the plant known as the yield components cumulatively determine the overall yield. In soybean, components such as pod number, seed weight, seed number, and population density among others contribute to the overall yield and an increase in these components may significantly influence the yield. Enhancing these components and subsequently, the overall yield requires the adoption of various agronomic practices such as the application of plant growth regulators.

Numerous researches have studied the influence of PGRs on these components and their influence on the overall yield of plants. Applying NAA revealed an increment in the yield of soybean (*Basuchaudhuri, 2016*). Seed yield, pod number per plant, and overall yield of soybean increased after NAA application (between 10–100 ppm; *Dhakne et al., 2015*). A higher pod and seed number per plant were also recorded in soybean treated with 250 ppm of CCC resulting in higher seed yield (*Kumar et al., 2002*). Soaking seeds

with 10 ppm gibberellic acid before sowing, then spraying again at the vegetative and flowering stage recorded a significant increase in yield (2.62 tons per hectare; *Domingo, 1981*). Soybean plants treated with 300 mg/l of BA decreased pod abortion in the lower, middle, and upper third of the canopy resulting in high productivity (*Borges, Junior & De Freitas Santos, 2014*). BAP application to racemes before efflorescence, reduced flower and pod number whereas application around 7 d after efflorescence considerably improved the rate of pod set (*Nonokawa et al., 2007*). The time of application can affect the productivity of the soybean plant, thus application during the end of flowering is more advantageous for soybean cultivation.

A study involving the foliar application of Tri-iodo benzoic acid (TIBA) to soybean plants revealed an enhanced pod number per node (*Noodén & Nooden, 1985*). Also, *Kumar et al. (2002)* detected a substantial increase in the pod and seed number per plant when 50 ppm of TIBA was applied and consequently increased the grain yield in soybean. Treatment with 15 to 120 ppm TIBA facilitated fruit development and better conditions for podding and also increased seed yield. However, higher levels were injurious to the plant (*Buzzello et al., 2013*). Again, seed yield, weight, and harvest index were observed to increase in soybean cv. Harit Soya when treated with 20 ppm $GA_3$ at bud initiation and 50% flowering stage (*Upadhyay & Rajeev, 2015*). Higher concentrations of $GA_3$ at 100 ppm efficiently augmented the number of pods and seeds per plant, seed weight per plant, and overall seed yield of soybean (*Sarkar, Haque & Abdul Karim, 2002*). Soybean cv. JS-9305 saw a significant increase in the seed and pod number per plant, seed weight, and total yield when seeds were primed (*Agawane & Parhe, 2015*). Different concentrations of $GA_3$ significantly influenced the pod number per plant, seeds per pod, 1,000 seed weight, and economic yield of soybean (*Dhakne et al., 2015*).

*Nagel et al. (2001)* revealed that the exogenous application of cytokinin improved the total yield of soybean by increasing the pod and seed number per plant. Total seed production increased with increasing cytokinin concentration, suggesting that the total yield of soybean may depend on cytokinin levels. Foliar application of plant growth regulators at both R1 and R3 stages decreased flower drop and efficiently influenced the transport of assimilates from source to sink. This, however, increased the pod number per plant significantly in many crops (*Copur, Demirel & Karakuş, 2010*). In soybean, applying $GA_3$ at the R1 and R3 stages significantly increased the pod number per plant.

Also, PGRs increased pod length which determines the seed per pod of soybean plants, depending on the time of application. For instance, the application of salicylic acid at the R1 stage and $GA_3$(foliar) at the R1+R3 stages recorded the highest pod length in soybean compared to other treatments applied at different stages of growth (*Khatun et al., 2016*). According to *Dhankhar & Singh (2009)*, applying $GA_3$ increased the pod length of the soybean plants. The seed number per pod increased with the application of salicylic acid and Kn spray at vegetative, R3, and R1+R3 stages which were the same for the foliar application of $GA_3$ at the R1+R3 stage. The application of salicylic acid and 100 ppm $GA_3$ had the best effect. Similarly, *Devi et al. (2011)* recorded an increase in the seed number of soybean with the application of 50 ppm salicylic acid at the R1+R3 stage. Foliar spray with 50 ppm salicylic acid and 200 ppm of ethrel applied at R1, R2, and R1+R3 growth stages in

soybean cv. JS 335 increased seed yield, attributed to better vegetative growth (*Devi et al., 2011*).

Further researches have reported that the application of PGRs increases the seed weight of soybean. *Zholobak (1986)* recorded a 20% increase in 100-seed weight when Kn was applied compared to the control. Similarly, *Khatun et al. (2016)* recorded the highest 100-seed weight in soybean with the application of salicylic acid and Kn spray at the R1+R3 and R3 stages respectively. Similar results were recorded for the foliar application of Kn at the vegetative stage. *Devi et al. (2011)* also reported an increase in the 100-seed weight of soybean with the application of Kn (500 ppm) at the R1+R3 stage. Kn application (40ppm) also increased the total pod set per plant, net return, and yield of soybean respectively with high concentrations recording the highest yield (*Passos et al., 2011*). This increase is attributed to the ability of plant growth regulators to improve photosynthesis and enhance the translocation of photosynthates in reproductive sinks of soybean plants which increased the cumulative effect of yield components and subsequently, increased the overall yield of soybean. However, the time of application of the growth regulators significantly influenced the 100-seed weight of soybean.

These results project that flower and pod dropping of soybean can be minimized by using exogenous BA, GA$_3$, Kn, and SA. Also, these PGRs can influence the diverse yield components which intend to affect the overall yield by restricting stress impacts. Under various environmental stresses, the application of PGRs may reduce the effects of these stresses on growth and overall yield. For instance, under salt stress, foliar application of n-triacontanol increased the specific leaf area (SLA) by increasing the leaf osmotic potential which may be due to increased organic solutes ($Na^+$ and $Cl^-$) in the leaves (*Dhakne et al., 2015*). One detrimental effect of salt stress in plants is to induce a reduction in relative water content of leaves, leading to loss of turgor which makes available little water for cell extension processes (*Krishnan & Kumari, 2008*). However, the foliar application of n-triacontanol alleviated this effect and increased the SLA. Also, exogenous ABA increased soybean yield under water deficit conditions (*Zhang et al., 2004*).

## Quality

Mostly, the nutritional constituents of plants describe their quality. Research evidence suggests that plant growth regulators significantly increase the nutritional value/quality of soybean seeds. Seed quality parameters like protein content are of special concern to food-grade soybean production. Low protein content in soybean seeds is undesirable for soymilk as well as tofu yield. Soluble protein has essential functions in plant growth and is a key constituent of several plant enzymes reflecting the plant's metabolism in totality. Foliar applications of 100 mg/l salicylic acid, 10 mg/l paclobutrazol (PP$_{333}$), or 5 g/l humic acid to soybean plants had a positive influence by maximizing its nutritional value (*El-Aal & Eid, 2017*). Salicylic acid at 50 ppm increased the quality of soybean and yield components compared to the control (*Devi et al., 2011*). Treating plants with GA$_3$ reduced seed protein whiles GA$_3$ spray increased oil content (*Travaglia, Reinoso & Bottini, 2009*). Higher doses of NAA also resulted in larger seed size and high protein contents but lower oil contents in soybean (*Basuchaudhuri, 2016*). Exogenous cycocel application increased protein and oil

contents in soybean seeds (*Devi et al., 2011*). Although the application of PGRs influenced the nutritional value of the soybean plant, the time of application played an essential role. Application of GA₃ and salicylic acid (spray) significantly improved the protein percentage of soybean with the highest protein percentage recorded for GA₃ application at R1+R3 stage and salicylic acid (spray) at the vegetative, R1, and R3 stages (*Khatun et al., 2016*). Although GA₃ did not greatly influence the oil content, application during flowering and pod development showed a significant increase. The increase in oil yield correlated directly with grain yield of soybean (*Giri et al., 2018*). Furthermore, spraying salicylic acid on leaves and stems of soybean cv. BARI at the vegetative, R1, R3, and R1+R3 stages also increased the protein and moisture content of seeds (*Khatun et al., 2016*). In a recent study, foliar application of JA (0.5 mM) and SA (1 mM) enhanced the oil quality and yield as well as the fatty acids of soybean seeds by increasing the unsaturation index (UI), linolenic acid, and linoleic acid contents while reducing the oleic acid content. SA had the best effect on oil quality and yield both under salt toxicity and normal conditions (*Ghassemi-Golezani & Farhangi-Abriz, 2018*). Moreover, floral application of exogenous MeJA (0.1 mM) and SA (0.1 mM) increased the isoflavones content in soybean seeds (*Saini et al., 2013b*). Similarly, *Hamayun et al. (2015)* reported a significant increase in the isoflavones content of soybean with the exogenous application of Kn which enhanced their biosynthesis directly by upregulating the expression of key pathway genes. Taken together, PGRs have the potency of enhancing the quality of soybean seeds both under stressed and normal conditions.

## Effects on biotic stress

Biotic stress results from damages caused by living organisms such as bacteria, viruses, fungi, insects, nematodes, and weeds. These organisms subject crop plants to several kinds of infections and diseases which diminishes plant vigor and in severe conditions, death as a result of direct nutrient deprivation (*Gull, Lone & Wani, 2019*). Plants effectively combat the detrimental effects of biotic stresses through their inducible defense mechanisms, mostly controlled by phytohormones such as SA, JA, ABA, ET, and BR (*Koo, Heo & Choi, 2020*).

Exogenous phytohormone supplementation enhances plant defenses against pathogenic diseases. For instance, application of PGRs such as SA, IAA, IBA, NAA, GA₃, BAP, Kn, BTH (a synthetic SA analog), and 2, 4-D inhibit sporulation, mycelium growth, mycotoxin synthesis, germ tube elongation, toxin production, aflatoxin synthesis, pathogenicity, and growth of pathogens including *Aspergillus spp. (umbrosus, parasiticus,* and *nidulans), Nigrospora spp.* (*oryzae* and *sphaerica), Botrytis spp.* (*cinerea* and *allii), Penicillium expansum, Colletotrichum dematium, Alternaria alternata, Xanthomonas oryzae* pv. *oryzae* (*Xoo*), *Sclerotinia sclerotiorum* and *Magnaporthe oryzae* among others (*Abass, 2017*; *Bucio-Villalobos et al., 2005*; *Qin et al., 2003*). The growth inhibition reduced disease incidences such as mold disease in cucumber (*Al-Masri et al., 2002*), early blight in potatoes (*Michniewicz & Rozej, 1988*), botrytis blight in cut rose flowers (*Shaul, Elad & Zieslin, 1995*), and improved resistance to bacterial leaf blight and blast disease in rice (*Shimono et al., 2007*). Furthermore, exogenously applied JA, SA, ET, and Kn induced resistance to root-knot nematodes in rice by upregulating PR genes (*Nahar et al., 2011*)

and inhibited the penetration and growth of root-knot nematodes in tomatoes (*Dropkin, Helgeson & Upper, 1969*). In addition, 2,4-D was used as herbicide to effectively eradicate broad-leaved weeds, triggered by uncontrollable cell division leading to death.

In soybean, exogenously applied BTH and 1-aminocyclopropane-1-carboxylic acid (ACC, an ET analog) significantly induced resistance to *Phytophthora sojae* which causes the Phytophthora root and stem rot disease by enhancing the expression of PR genes (*Sugano et al., 2013*). Similarly, ACC treatment inhibited hyphal growth of *Phytophthora sojae* and suppressed disease incidence in excised hypocotyls and cotyledons, respectively (*Park et al., 2002*). *Dann et al. (1998)* reported reduced severity of white mold disease caused by *Sclerotinia sclerotiorum* after BTH application. Furthermore, exogenously applied MeJA decreased root-knot nematode (*Meloidogyne spp.*) infections by significantly reducing nematode population (*Hu et al., 2017*). Similarly, MeJA applications reduced soybean cyst nematode (*Heterodera glycines*) infections by upregulating the expression of (E, E)-a-farnesene synthase gene in soybean roots (*Lin et al., 2017*). Thus, exogenous JA might be an alternative strategy to induce soybean resistance against nematode infections. However, the actions of these growth regulators were concentration-dependent (*Al-Masri et al., 2002*).

## Effects on abiotic stress

Abiotic stresses such as salinity, drought, and severe temperatures negatively affect the morphology and physiology of plants by affecting the gene regulatory mechanism of cellular pathways (*Egamberdieva et al., 2017*). PGRs are chemical messengers that act in very small quantities to mediate abiotic stress response in plants. Exogenous application of these growth regulators offers an alternative approach to control abiotic stress which decreases plant growth and productivity (*Wani & Sah, 2014*). Several abiotic stresses including waterlogging, drought, low temperature, and salt stress negatively influence the growth and development of soybean plants, however, different growth regulators have successfully mitigated these stress factors leading to improved growth and productivity (Fig. 3).

### *Low temperature*

Low temperature, one of the major limiting factors of plant growth and productivity, is defined as a drop in temperature to a level low enough but not freezing (>0 °C) to influence cellular function, abnormalities at various levels of cell organization, and inhibit growth. Membrane damage, reduced cellular respiration, and enhanced ROS production are the characteristic effects of low temperature stress in plants (*Xing & Rajashekar, 2001*). In soybean, all growth and developmental stages (from sprouting to seed filling and maturity) are negatively affected by low temperature, making soybean a cold-sensitive plant (*Holmberg, 1973*).

Low temperature reduces germination rate in soybean by delaying germination, that is, the lower the temperature, the lesser the rate of germination. Several studies have reported the adverse effects of low temperature on pod set and seed yield in soybean due to major factors such as poor growth, abscission of flowers and pods, and inadequate seed filling (*Matsukawa, 1994*). For instance, low-temperature treatment before and during flowering

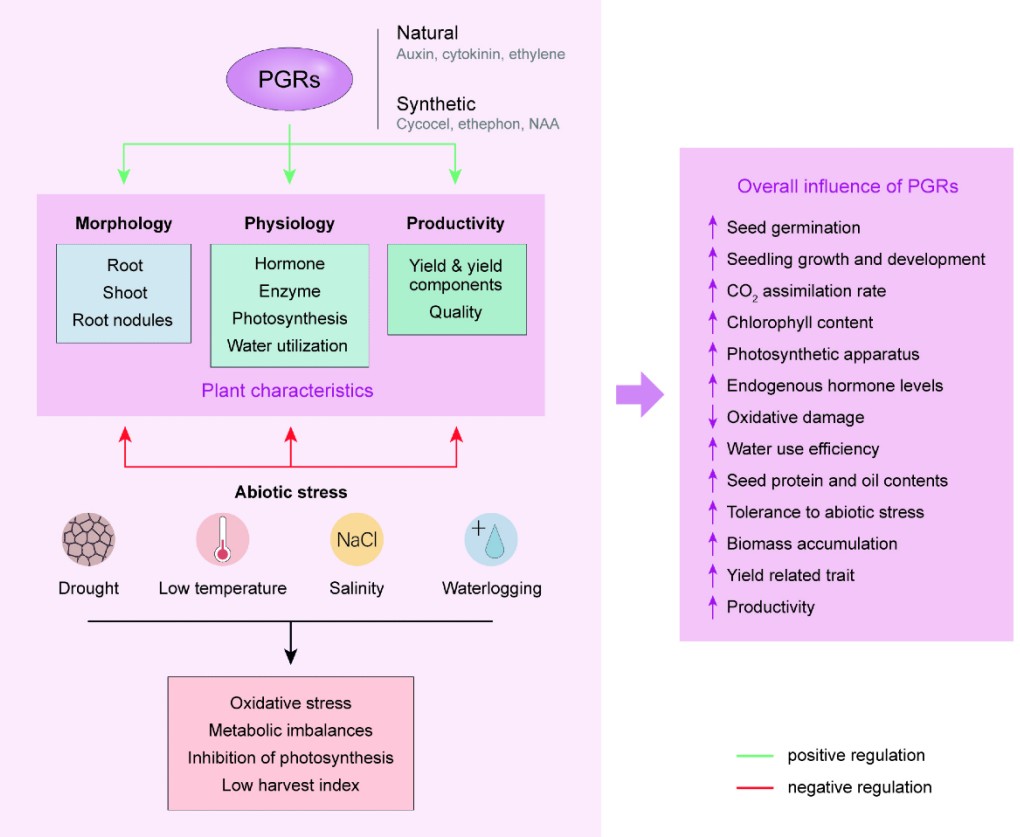

**Figure 3** Summary of the effects of plant growth regulators on the growth and development of soybean.

resulted in ruinous pod abscission in soybean (*Michailov, 1989*). *Funatsuki et al. (2004)* recorded a high reduction in soybean yield resulting from malformed pods and reduced seed filling upon exposure to low temperature before and during flowering. Furthermore, exposing soybean plants to low-temperature stress before flowering decreased the rate of fertilization and pod set marked by a decline in male fertility (*Goto, 1972*), attributed to the abnormal development of pollen grains, decreased pollen dispersal, and reduced number of pollen grains on the stigma (*Ohnishi, Miyoshi & Shirai, 2010*). According to *Kurosaki & Yumoto (2003)*, long-term exposure of soybean to low temperature diminished pod set caused by a decline in the number of fertilized flowers which severely influenced grain yield. Similarly, *Ohnishi, Miyoshi & Shirai (2010)* revealed that low-temperature stress significantly decreased pod set at the flowering stage (12.5 d and 3-4 d before anthesis) by disrupting normal development of pollen grains, pollen dispersion, reduced number of pollen grains deposited on stigma, thus reducing pollination. However, the response to low temperature at the flowering stage and pod setting ability upon exposure to low-temperature stress is cultivar dependent, mainly based on the tolerance level (*Kurosaki & Yumoto, 2003*).

To eliminate the detrimental effects of low temperature on the growth and productivity of soybean and further improve tolerance to low temperature, plant growth regulators have been demonstrated through several studies as an effective agronomic technique. In soybean, tolerance to low temperatures has been achieved through the exogenous application of PGRs, and this promoted growth and seed yield. For instance, low concentrations of 5-aminolevulinic acid (ALA) induced low-temperature tolerance in soybean plants by improving heme-catabolism and antioxidant enzyme (heme-catalase (CAT) and heme-oxygenase-1 (HO-1)) activity. These enzymes inhibited oxidative damage by scavenging ROS, thereby protecting soybean plants from the ravaging influence of low-temperature stress. Pre-treatment with 5 µM of 5-aminolevulinic acid before cold treatment had the best effect (*Balestrasse et al., 2010*). Based on the results outlined above, it can then be proposed that the exogenous application of PGRs enhance soybean tolerance to low temperature and may be useful in preventing yield losses due to low temperatures.

### Salt stress

Salt stress as an abiotic stress factor embraces all salt-related problems emanating from the excessive supply of NaCl (sodium chloride) either by natural accrual or artificial supply through irrigation (*Flowers & Flowers, 2005*). Salt stress has overwhelming effects on the overall growth and development of plants by affecting numerous vital physiological processes in plants (*El Sayed & El Sayed, 2011*) like protein synthesis, photosynthesis, and energy metabolism (*Ahmad et al., 2019*). Generally, salt stress interferes with complex hormone interaction, ion toxicity, osmotic effects, and nutritional balances in plants (*Ahmad et al., 2019*).

In soybean, salt stress disrupts overall growth and development (*Phang, Shao & Lam, 2008*) by influencing photosynthesis, seed germination, nutritional imbalance, nutrient uptake, and productivity (*Zhu, 2016*). For instance, *Shu et al. (2017)* indicated that salt stress impedes germination by dwindling the GA/ABA ratio in soybean. Similarly, *Kumar (2017)* indicated a decrease in germination and seedling growth in soybean under salt stress. *Khan et al. (2016)* and *Ghassemi-Golezani et al. (2010)* reported a significant reduction in all yield-related traits under salt stress, eventually leading to significant decrease in final yield of soybean. *Ruzhen & Yiwu (1994)* recorded nearly 40% yield loss in soybean as a result of high soil salinity. A recent study reported a decline in the amount of grain oil and seed protein of soybean under salt stress. In contrast to the control, the oil and protein contents diminished as the salt concentration escalated (*Ghassemi-Golezani et al., 2010*). In another study, *Egbichi et al. (2013)* reported that exposing soybean plants to salt stress-induced oxidative damage of membrane lipids in root nodules.

Several techniques such as conventional breeding and exogenous application of PGRs have been reported to enhance plants' tolerance to salt stress, promote plant growth, and consequently increase productivity (*Javid et al., 2011*). Though soybean plants are adapted to several mechanisms that induce resistance to salt stress, agronomic techniques to eliminate salt stress in soybean are minimal. Plant growth regulators offer a lasting solution to the many growth challenges faced by soybean plants under salt stress. The exogenous application of fluridone (FLUN) promoted seed germination of soybean under

salt stress by inhibiting ABA biosynthesis and promoting GA biosynthesis (*Shu et al., 2017*). Exogenously applied Kn ameliorates the adverse effects of salt stress by declining ABA levels (growth inhibitory hormone) and elevating GAs and SA (growth and defense hormones) levels. These endogenous hormone fluctuations caused by the influence of exogenous Kn, significantly increased fresh weight, plant height, and dry weight of soybean plants thereby promoting growth through enhanced chlorophyll content and leaf area (*Hamayun et al., 2015*). In another study, the combined effect of exogenous SA and SNP, a nitric oxide (NO) donor significantly improved shoot biomass (both fresh and dry), and the leaf area of soybean plants under salt stress by increasing the chlorophyll content. This stimulated the photosynthetic ability of the soybean plants and subsequently improved growth. However, 100 mM salicylic acid and/or 100 $\mu$M sodium nitroprusside had the best effect (*Simaei et al., 2011*).

Plant growth regulators, either single or combined can alleviate salt-induced oxidative damage and confer resistance to salt stress (*Dong et al., 2017*). According to *Krishnan & Kumari (2008)*, foliar spray of n-triacontanol (TRIA, a saturated primary alcohol) reduced the harmful effects of salt stress on soybean plants and promoted growth. This was evident with the observed increase in all growth parameters (fresh weight, dry weight, root and shoot length), physiological parameters (relative water content, SLA, and leaf weight ratio), and biochemical parameters (total soluble sugars, soluble proteins, nucleic acids (DNA and RNA) and chlorophyll content) tested. The accelerated growth resulted from enhanced cell division and expansion (*Sun et al., 2016*), increased nutrient content ($K^+$ and $Ca^{2+}$) and its metabolism which enhanced the absorption of $Na^+$ ions (*Song et al., 2016*), increased nitrogen availability (*Parashar & Verma, 1993*) and decreased chlorophyllase and protease activity (*Dong et al., 2017*). Thus, suggesting the ability of n-triacontanol to restore normal metabolism in salt-stressed soybean plants. Although growth was accelerated, there was a decline in proline accumulation and leaf osmotic potential. From the above report, it is speculated that plant growth regulators can ameliorate the adverse effects of salt stress on soybean plants and accelerate growth, thereby increasing productivity.

### Drought

Drought stress is characterized by low atmospheric and soil humidity as well as high ambient temperature influenced by an imbalance between water intake from the soil and evapotranspiration oscillations (*Lipiec et al., 2013*). Drought stress is detrimental to the growth and development of plants by influencing their biochemical, morphological, physiological, and genetic resources which diminishes productivity (*Khan & Mazid, 2018*). Thus, drought hinders the proper functioning of plants by interrupting the water use potential and turgidity. Moreover, drought influences oxidative damage which may degrade proteins, lipids, nucleic acids, and cause lipid peroxidation in plants (*Lipiec et al., 2013*).

Soybean is a drought-sensitive crop and its exposure to drought stress particularly during seed filling (*Desclaux, Huynh & Roumet, 2000*), flowering, and pod-filling stages (*Kobraee, Shamsi & Rasekhi, 2011*) reduces photosynthesis, thus, decreasing yield (*Liu, Jensen & Andersen, 2004*). During drought stress, the rate of abortion at the early pod-filling stage

increases in soybean and thus affect productivity (*Lie, Anderson & Jensen, 2003*). In soybean, water stress including drought stress approximately caused a 40–60% yield loss (*Ahmed et al., 2013*). *Shou et al. (1991)* recorded a 46% and 20% decrease in soybean yield at the flowering and seedling stages respectively, due to decreased photosynthesis under water stress.

In an attempt to increase soybean growth and productivity under drought conditions, many drought-tolerant soybean varieties have been developed through conventional and molecular breeding techniques. Another optional approach by researchers to effectively promote plant growth and increase drought tolerance is the application of growth regulators. Exogenous application of PGRs such as abscisic acid, brassinolide, and uniconazole under drought conditions increases soybean tolerance to drought stress and improves productivity. For instance, uniconazole application under drought stress improved soybean tolerance to drought which increased biomass accumulation and seed yield (*Zhang et al., 2007*). The higher yields recorded in uniconazole-treated soybean plants may be as a result of the enhanced transport of $^{14}$C assimilates from leaves to pods. *Zhang et al. (2008)* indicated that treatment with BL before drought stress minimized water-deficit yield loss in soybean. In another study, foliar application of MeJA eased drought stress in soybean plants thereby improving drought tolerance and yield in comparison to unstressed plants (*Mohamed & Latif, 2017*). The drought tolerance exhibited by soybean plants treated with MeJA (foliar spray) resulted from the elevated levels of soluble sugars and secondary metabolites including flavonoids and phenolic compounds associated with plant defense response against several biotic and abiotic stresses (*Kim et al., 2007*).

This implies that exogenous PGRs mitigate drought stress and improve drought tolerance in soybean through increased water use potential, leaf water potential, nitrate reductase activity, chlorophyll contents, and photosynthesis (*Zhang et al., 2007*), contributing to increased translocation of $^{14}$C assimilates, dry weight accumulation, and high leaf area (*Mohamed & Latif, 2017*). This significantly increased the yield-related parameters and subsequently improved the overall seed yield of soybean, indicating the ability of PGRs to stimulate soybean growth and improve yield even under drought stress.

### Waterlogging

Waterlogging occurs when plant roots are surrounded by water or the shoots are partially or entirely submerged (*Sullivan et al., 2001*). Waterlogging is the most frequent flooding condition, characterized by excessive rainfall and/or the overspill of rivers (*Kim et al., 2015*) and it is one of the limiting factors in global soybean (a flood-sensitive crop) cultivation. Waterlogging negatively influences many physiological processes in plants by reducing photosynthesis, nutrient uptake, and causes hormonal imbalance that negatively affects the growth and yield of soybean (*Kim et al., 2015*; *Valliyodan et al., 2017*). For example, substantial yield loss was observed in soybean under severe waterlogging stress during the vegetative (17 to 43%; *Reyna et al., 2003*) and reproductive (50 to 56%; *Scott et al., 1990*) growth stages. Similarly, soybean plants exposed to waterlogging stress recorded a 17% and 57% yield loss at the vegetative and reproductive stages respectively, compared to non-stressed plants (*Nguyen et al., 2012*). A recent report by *Rhine et al. (2010)* suggested
a 39% yield loss in waterlogging-tolerant soybean varieties as against the 77% yield loss in susceptible varieties under severe waterlogging conditions.

Plants develop many strategies to mitigate the adverse effects of waterlogging stress which improves their tolerance. Among them, is the escape strategy which allows gaseous exchange between cells and their environment due to the changes that occur in both the anatomical and morphological structures of the plant including adventitious root formation, shoot elongation, and aerenchyma cell development (*Bailey-Serres & Voesenek, 2008*). Recently, several studies have reported the use of exogenous plant growth regulators as an effective alternative to mitigate the deleterious effects of waterlogging stress in plants. *Kim et al. (2015)* indicated that the application of GA, and ETP improved tolerance to waterlogging stress than in control plants. Treatments 50 µM and 100 µM ETP had the best effect, recording the highest GA concentrations even after short-term stress. The results revealed that GA- and ETP-treated soybean plants developed an escape strategy under waterlogging by the hyper-elongation of the shoot which increased the overall plant height. This then confirms that the accumulation and fluctuations in the endogenous GA levels induce the escape strategy. In another study, *Kim et al. (2018)* reported the ability of ETP to ameliorate waterlogging stress in soybean plants. This ameliorative effect of ETP was due to its ability to induce the overexpression of antioxidant enzymes. Overall, this could promote growth and improve soybean productivity under waterlogging stress.

Root length is among the key response indicators to waterlogging stress in plants. Tolerance of soybean plants to waterlogging stress correlates sturdily to the overall root growth (length, surface area, and dry weight; *Sallam & Scott, 1987*). Recently, *Kim et al. (2018)* documented that waterlogging stress can impede root growth in soybean by reducing the root size and RSA. However, the exogenous application of ETP to soybean plants subjected to waterlogging stress increased root size and RSA which promoted root growth and nutrient (K and P) uptake.

ABA is a key hormone in water stress response. According to recent research, endogenous ABA contributes to the development of aerenchyma cells in root tips during waterlogging, thus facilitating the absorption and transport of oxygen (*Shimamura et al., 2014*). Endogenous ABA decreases in response to waterlogging stress to keep the stomata open, providing greater surface area for oxygen entry and removal of excess water (*Shimamura et al., 2014*). Increasing research suggests that the downregulation of endogenous ABA suppresses suberin biosynthesis which keeps the root cell unsuberized for aerenchyma cell development in soybean. Similarly, *Kim et al. (2015)* observed a significant decrease in ABA contents and well-developed aerenchyma cells in waterlogging tolerant soybean variety compared to non-tolerant variety and control. This suggests the crucial role of ABA in regulating soybean tolerance to waterlogging stress, thus promoting growth and productivity. In conclusion, the exogenous application of plant growth regulators can mitigate waterlogging stress in soybean by improving waterlogging tolerance through the development of the escape strategy, increased accumulation of endogenous GAs, induced antioxidant enzyme activity, and increased root growth which promoted nutrient uptake. This in effect promoted the overall growth and productivity of soybean under waterlogging stress. However, the significant internode growth observed in the GA-
and ETP-treated soybean plants subjected the plants to lodging (*Seo et al., 2017*). We can therefore propose that the use of exogenous GA and ETP may be ineffective for commercial soybean production.

## CONCLUSION

PGRs are commonly applied in agriculture to improve germination, seedling establishment, growth, and development as well as yield even under unfavorable environmental and soil conditions. At low concentrations, PGRs control the structure and function of the cell, cell division and expansion together with the regulation of environmental stresses. Application of PGRs directly to the leaves, shoots, buds, roots, and flowers protects plants from biotic and abiotic stresses, enhances water use efficiency, breaks dormancy, and improves drought tolerance among others. PGRs applied at the proper period of growth and in appropriate concentrations, influence the yield and yield components of soybean (as outlined in Fig. 3).

Several studies only concentrated on the essential roles of PGRs on various physiological, morphological, and biochemical properties together with yield and quality traits of soybean. Nevertheless, studies on the use of PGRs to enhance growth and development of the anatomical structures that contribute to the entire well-being of the plant and consequently its yield is inadequate. Therefore, further studies are needed to bridge this aforementioned gap. Future works could be focused on the effects of plant growth regulators on these structures to increase our understanding of the mechanisms employed by various growth regulators to influence growth of these plant parts and increase productivity. All in all, one could judge that PGRs do not only promote the growth and development of the soybean plant but also increase productivity. An illustration of the different growth stages of the soybean plant, from germination to maturity is presented in Fig. 4.

**Abbreviations**

| | |
|---|---|
| **PGRs** | Plant growth regulators |
| **IAA** | Indole acetic acid |
| **ABA** | Abscisic acid |
| **GA** | Gibberellins |
| **ETP** | Ethephon |
| **JA** | Jasmonic acid |
| **DTA-6** | Diethyl aminoethyl hexanoate |
| **Rubisco** | Ribulose-1,5 bisphosphate carboxylase |
| **MBTA** | Diethyl-2-(4-methylbenzoxy) ethylamine |
| **ROS** | Reactive oxygen species |
| **GST** | Glutathione S-transferase |
| **NAA** | Naphthalene acetic acid |
| **DCPTA** | 2-(3,4-dichlorophenoxy) trimethylamine |
| **GSH** | Glutathione |
| **ET** | Ethylene |
| **SA** | Salicylic acid |
| **CCC** | Chlormequat-chloride/2-chloroethyltrimethylammonium chloride |

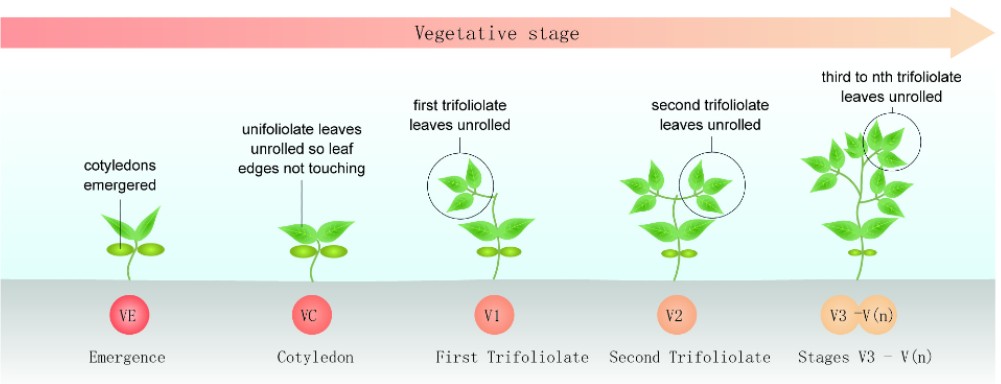

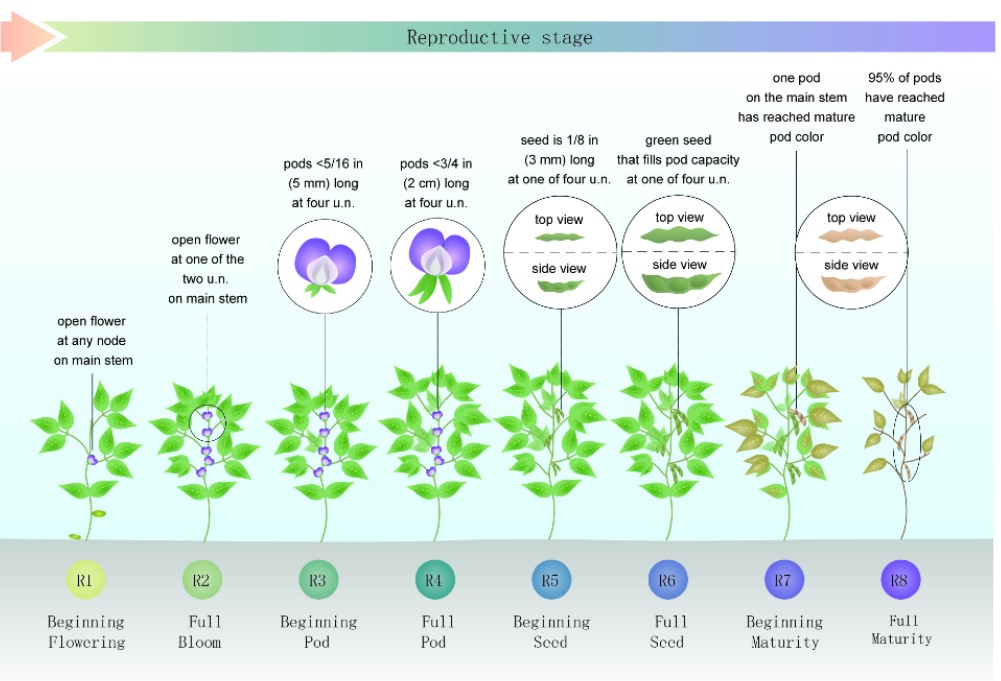

u.n. : uppermost nodes

**Figure 4** **Stages of soybean growth from germination to maturity.** *Source* modified from *Hodgson et al. (2012)*.

| | |
|---|---|
| **TIBA** | Tri-iodo benzoic acid |
| **BMVE** | 2-(N-methylbenzylaminoethyl)-3-methylbutanoate |
| **BA** | Benzyladenine |
| **Kn** | Kinetin |
| **BL** | Brassinolide |
| **BAP** | 6-Benzylaminopurine |
| **PEPCase** | Phosphoenolpyruvate carboxylase |
| **RSA** | Root surface area |

| IBA | Indole butyric acid |
|-----|---------------------|
| EBL | 24-epibrassinolide |
| GR | Glutathione reductase |
| WUE | Water use efficiency |

## ACKNOWLEDGEMENTS

The authors are thankful to Ms. Linda Adzigbli for the critical review and useful suggestions during the manuscript preparation.

### Funding

This work was supported by the General Program of National Natural Science Foundation of China (31871576), the Project of Enhancing School with Innovation of Guangdong Ocean University (230420006), and the Studies on Resistance Resources and Molecular Mechanisms of Sweet potato Weevil in South China (U1701234). The funders had no role in study design, data collection and analysis, decision to publish, or preparation of the manuscript.

### Grant Disclosures

The following grant information was disclosed by the authors:
General Program of National Natural Science Foundation of China: 31871576.
Project of Enhancing School with Innovation of Guangdong Ocean University: 230420006.
Studies on Resistance Resources and Molecular Mechanisms of Sweet potato Weevil in South China: U1701234.

### Competing Interests

The authors declare there are no competing interests.

### Author Contributions

- Hanna Amoanimaa-Dede conceived and designed the experiments, performed the experiments, analyzed the data, prepared figures and/or tables, authored or reviewed drafts of the paper, and approved the final draft.
- Chuntao Su and Hang Zhou analyzed the data, authored or reviewed drafts of the paper, and approved the final draft.
- Akwasi Yeboah analyzed the data, prepared figures and/or tables, authored or reviewed drafts of the paper, and approved the final draft.
- Dianfeng Zheng and Hongbo Zhu conceived and designed the experiments, authored or reviewed drafts of the paper, and approved the final draft.

### Data Availability

This is a literature review and there is no raw data.

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
