# Peer review of "Growth regulators promote soybean productivity: a review"

_PeerJ, doi:10.7717/peerj.12556_

## Round 0.1 · original submission · Major Revisions

Please revise the manuscript as per the reviewers' comments.

·

Basic reporting

Basic reporting: The manuscript well described the role of different growth regulators on the productivity of different crops. The language is clear. Though the importance of soybean is mentioned well, however, little more emphasis on the role of growth regulators on soybean productivity should be given in the Introduction Section to make it more meaningful of targeting soybean as a crop in the review.

Experimental design

Study design: Followed standard procedures for presenting the information as a review article.

Validity of the findings

Validity of the findings: Some suggestions are given to improve the manuscript such as-
i. Line 53 – 54: “This has prompted its consumption as a staple food…… quality”. The term staple is not much appropriate for soybean as a food crop. Better to revise the sentence.
ii. Line 58 – 59: “Generally grain legumes are excellent source of energy but …………… still under rainfed”. The statement is not clear for the general readers, please make clear what do you meant for!
iii. Line 76 – 78. “In soybean production, higher yields obtained …………….timely germination of seeds and uniform emergence of seedlings from the soil”. Timely germination and uniform emergence of course important criteria, but the yield is not much attributed to the two criteria as those are not direct yield contributing characters. Please revise the sentence.

iv. Line 114 – 401: While discussing the general characteristics and role on crops of different growth regulators, a very little reference of soybean has been mentioned, though that for other crops are well described. More emphasis is required to place on soybean in this section.

References: Need serious improvement!! Just for example-
i. L 1099 – 1101: There is no volume or issue number of the journal.
ii. L 1102: It is mentioned Ahmad W et al. (2019). However, all the authors name should be mentioned. Similarly, L 1115. Many more also!!!!
iii. L 1157: chemical ecology should be Chemical Ecology; Line 1164: Trends in plant science should be Plant Science!! Many more of similar cases.

Additional comments

General comments: The manuscript is suitable for publication in PeerJ as a review paper after making necessary improvement as indicated.

Reviewer 2 ·

Basic reporting

The message of the review article is clear and easy to follow. However, the article could benefit from clarity in the writing style. The structures of the review should also be revised to elaborate more on the effect of PGRs with biotic environmental stress, especially with the majority of plant hormones being related to biotic stress that the authors mentioned in the review.

I have a concern regarding the usefulness of figures in this manuscript. In general, the figures are actually a list of the sections addressed in this review (accept Figure 2). I suggest that the authors prepare themselves, or work with a scientific illustrator to summary the different effects and targets of PGRs in graphic illustrations that is easy to understand. A good place to start is lifeology.io where you can connect with illustrators who can help on this.

Experimental design

The literature survey method is clear, sufficient and easy to replicate.

Validity of the findings

no comment as this is a literature review article

Additional comments

The review is a thorough report of plant growth regulators on soybean production, reflected by the enhancement of crop quality, plant growth & development, abiotic stress tolerance. I commend the authors on the comprehensiveness of this literature survey. Although the review covered most of the aspects of PGRs in soybean production laid out in the abstract and introduction, there are several concerns that need to be addressed before this manuscript can be accepted.

1. The review mentioned extensively about abiotic stress, while a lot of plant growth regulators mentioned in the manuscript is actually also related to biotic stress responses (JA, MeJA, SA, ABA, Polyamines, etc.). Section 3.1.3 also mentioned root nodulation by rhizobia, therefore, it would be of readers’ special interest if the authors could include a separate section to briefly discuss the effect of PGRs on biotic interactions and how they would influence soybean yield.

2. Line 53: A summary table of soybean nutritional value would be very useful
3. Line 80: please spell out “m” as “million”. I also suggest the authors to remove this methodology section as this is a quite standard approach in surveying the literature for review articles. In addition, I assume that all the surveyed studies were already cited, so there is no need to mention the amount of hundreds of publications retrieved in the search here.

4. Line 124: what do you mean by “repair”? Can the authors provide an example of this effect as it is the only place where this effect was mentioned?

5. Line 150-153: What is the purpose of mentioning anti-auxin compounds here? What are their significances and applications?

6. Line 164: I suggest the author change “number of leaves per plant” to “leaf number”

7. Line 195-203: Same point as #6. I understand that the focus of this review is plant growth enhancement (as stated in line 28-30: “In an attempt to expatiate on the topic, current knowledge, and possible applications of plant growth regulators that improve growth and yield have been reviewed and discussed”). The functions of GA inhibitors as plant growth inhibitors, therefore, are not relevant to the focus of this manuscript.

8. Line 568: please spell out enzyme names if they are mentioned for the first time in the manuscript

9. Line 698-700: the authors mentioned R1 and R5 stage without mentioning what these growth stages were. Please define all the growth stages, and use the definitions consistently for other places in the manuscript (line 719: flower-initiation, pod-initiation, flower initiation + pod-initiation). An illustration of different growth stages would be very informative.

10. Line 722-725: These sentences are confusing. I suggest the authors re-write it to clearly convey the message coming from the results of the research cited.

11. Line 749: did the authors mean “flowering”?

12. Line 751: it is not clear what the authors meant by “keen attention”

13. Line 885: can the author elaborate on the relationship between JA and SA cited from Kim et al. 2007 and Wildermuth 2006 here. It is common knowledge that JA and SA pathways antagonize each other, so it would not make much sense when MeJA treatment could increase the production of SA. In addition, SA is a defense hormone and which can strongly inhibit plant growth and development. Therefore, one would expect that Sa treatment could reduce and not enhance yield due to abiotic stress. It would be great if the author can provide a clearer interpretation of these conflicting findings.

14. Line 932: Which group of PGRs does 5-aminolevulinic acid fall in to?


Some minor points:
1. The authors should include an index page so that readers can capture the big picture and framework of this review.

2. Italicize scientific names (Arabidopsis, Botrytis cinerea, Phakopsora pachyrhizi, Raphanus sativus L. var. longipinnatus Bailey, Glycine max (L.), etc.)

3. I suggest that the authors place the Abbreviation list at the beginning of the article so that readers can easily refer to it.

4. Although the majority of the manuscript is easy to follow, this manuscript can benefit greatly from some transparency in the writing style. I suggest that the authors work with the editor for some suggestions on editing services before re-submission of this manuscript.

·

Basic reporting

The manuscript submitted by Dede et al., is an interesting article. In this regard, the authors present an updated vision regarding the different investigations that demonstrate the effect of growth regulators on Soybean productivity. I believe that the manuscript will be of interest to researchers in the field.

Experimental design

Although, the manuscript is reasonably well written and has an adequate structure, the content in some sections looks superficial and lacks the critical insights that would make it a useful addition to the field.

Validity of the findings

No comment

Additional comments

Comments to the authors

Major comments:
1. Section 2.0, describes the role of major growth regulators on plant growth and development. The section is lengthy and just lists the general functions of plant growth regulators and whether they add anything to the article needs to be considered carefully. This section needs to be shortened substantially by highlighting the role of plant growth regulators in improving the Soybean productivity. Further, authors should include information about the synthetic or chemical growth regulators, which also play a major role in Soybean growth.

2. Table 1 reported the effects of different PGRs on growth and development of plants in general. Whether all the effects mentioned in the Table is applicable to Soybean growth and development is questionable. So, authors should change the Table 1 by highlighting the effects that are specific to Soybean.

3. Cycocel is an antagonist of GA3. In lines 470-473, authors mentioned the role of Cycocel in retaining more leaves per plant and attributed the effect to various other factors but not to GA3. Authors should clearly justify this in the text.

4. Line 474-475, authors mentioned the importance to Cytokinin in shoot development of plants. However, it’s role in Soybean shoot development is not reported in the section.

5. Recently, it showed that different concentration of cytokinin’s have large impact on Soybean nodulation (Celine Mens et al., 2018, Frontiers in Plant Science). Authors should include that in the section 3.1.3.

6. Full names of the abbreviated enzymes should be mentioned at first appearance in the text. For example, in the lines 556-557 the full forms of POD, CAT, SOD and GST are not mentioned.

7. Jasmonic acid and Brassinosteroids are well known for their antioxidant capacity (ASA-GSH cycle). Refer Seteiwy et al., 2020, Journal of science of food and agriculture and Hasan et al., 2020, Journal of plant interactions. I suggest the authors to include the importance of those growth regulators in the section 3.2.2.

8. Cross talk or interactions between various plant hormones has great effect on photosynthesis in plants. Authors should comment on the role of hormonal interaction on the rate photosynthesis in Soybean (refer Muller maren et al., 2021, Plant physiology). Further, Salicylic acid plays a major role in increasing the levels of enzymes (RUBISCO) and stability of subunits (PSII, OEE1 and OEE2) involved in photosynthesis (refer, Sharma et al., 2018 and 2019).

9. In many countries, Soybean is used as source of nutritionally important fatty acids. In section 3.3.2 authors should mention the effects PGRs on the quality of oil and fatty acids of Soybean seeds (refer Golezani-Ghassemi et al., 2018, Russian journal of Physiology).

10. ABA is a key hormone in water stress response. Shimamura et al., 2014, 2016 and Kim et al., 2015 showed that ABA negatively correlated with water logging tolerance. I suggested the authors to include the role of ABA in water logging tolerance in Section 3.44.

11. Figure 2 looks generalized representation of antioxidant defense system. I suggest the authors to modify the figure 2, highlighting the examples of specific PGRs on each antioxidant pathway/reaction in relation to Soybean.

Minor comments:

1. Format the references according to Journal guidelines. For example, Scientific names of the plants should be in Italics-line 1097, 1103 etc. The first letter of each word in the journal name should be written in uppercase-Line 1092, 1122 etc., Abbreviated journal names should be changed to Full form.

---

## Round 0.2 · Minor Revisions

The authors have revised the manuscript based on the reviewers' comments. However, it still needs some improvements. There are too many references cited in this paper (366) which are too much for such a review. I suggest reducing them. Cite only recent and relevant references. Some of the references are too old, especially in the tables. Please remove them. For ROS regulation and antioxidants, many of the recent references are not cited. Please check thrughly.
For Table 1, no need to create a separate column for units. Just put units adjacent to the values in the same column. Also, capitalize consistently e.g. Omega 3.

·

Basic reporting

no comment

Experimental design

no comment

Validity of the findings

no comment

Additional comments

The authors have addressed most of my concerns and I recommend the manuscript for publication in PeerJ.

---

## Round 0.3 · accepted · Accept

The authors properly revised the manuscript.